



# Observing river stages using unmanned aerial vehicles

Tomasz Niedzielski[1], Matylda Witek[1], and Waldemar Spallek[1]

[1]Department of Geoinformatics and Cartography, Institute of Geography and Regional Development, Faculty of Earth Science and Environmental Management, University of Wrocław, pl. Uniwersytecki 1, 50-137 Wrocław, Poland

*Correspondence to:* Tomasz Niedzielski (tomasz.niedzielski@uwr.edu.pl)

**Abstract.** We elaborated a new method for observing water surface areas and river stages using unmanned aerial vehicles (UAVs). It is based on processing multitemporal $m$ orthophotomaps produced from the UAV-taken visual-light photographs of $n$ sites of the river, acquired with a sufficient overlap in each part. Water surface areas are calculated in the first place, and subsequently expressed as fractions of total areas of water-covered terrain at a given site of the river recorded on $m$ dates. The logarithms of the fractions are later calculated, producing $m$ samples of size $n$. In order to detect statistically significant increments of water surface areas between two orthophotomaps we apply the asymptotic and bootstrapped versions of the Student's t-test, preceded by other tests that aim to check model assumptions. The procedure is applied to five orthophotomaps covering nine sites of the Ścinawka river (SW Poland). The data have been acquired during the experimental campaign, at which flight settings were kept unchanged over nearly 3 years (2012–2014). We have found that it is possible to detect transitions between water surface areas produced by all characteristic water levels (low, mean, intermediate and high stages). In addition, we infer that the identified transitions hold for characteristic river stages as well. In the experiment we detected all increments of water level: (1) from low stages to: mean, intermediate and high stages; (2) from mean stages to: intermediate and high stages; (3) from intermediate stages to high stages. Potential applications of the elaborated method include verification of hydrodynamic models and the associated predictions of high flows using on-demand UAV flights performed in near real-time as well as monitoring water levels of rivers in ungauged basins.

## 1 Introduction

A key problem in assessing skills of distributed hydrologic models, which predict water depth across a river channel and thus simulate flood extent, is access to up-to-date information on true inundation. There are numerous approaches used to carry out such observations. They include: terrestrial observations of flood damages carried out by volunteers who witnessed the flood, following the concept of volunteered geographic information (VGI) (e.g. Poser and Dransch, 2010), geomorphological survey and a subsequent mapping of landforms produced as a consequence of a high flow (e.g. Latocha and Parzóch, 2010), aerial photogrammetry (e.g. Yu and Lane, 2006a), use of satellite remote sensing (e.g Smith, 1997; Kouraev et al., 2004), application of airborne Light Detection and Ranging (LIDAR) measurements (Lang and McCarty, 2009) as well as use of photographs taken by unmanned aerial vehicles (UAVs) (Witek et al., 2014). However, there exist only several on-demand solutions which can serve a purpose of acquiring such data in real time (e.g. Schnebele et al., 2014). One of such attempts is the integration of





HydroProg, FloodMap and UAV, known hereinafter as HFU, which has been proposed by Niedzielski et al. (2015) after the initial feasibility study offered by Witek et al. (2014).

The HFU approach utilizes the UAV observations carried out in near real-time, i.e. when the integrated HydroProg (Niedzielski et al., 2014) and FloodMap (Yu and Lane, 2006a, b) solutions produce a real-time warning of predicted inundation. According to Niedzielski et al. (2015), the workflow of the HFU is the following: (1) HydroProg computes a hydrograph prediction for three hours into the future on a basis of multimodelling (being done routinely in real time with a pre-assumed frequency), (2) FloodMap uses the above-mentioned forecast as an input and enables mapping the hydrograph prognosis into the spatial domain (also being done routinely in real time with the same frequency), (3) the warning is distributed amongst the UAV team (to be done only when a number of inundated raster cells exceeds a certain threshold), (4) the UAV team carries out the survey in order to take aerial photographs of the river channel (not routinely, but only after the warning has been issued). It is known that hydrodynamical models may be incorrect, and this is also likely in the case of the HydroProg-FloodMap integration. The outputs from this integration are maps of predicted extent of terrain covered by water, known also as water surface area. Thus, in order to verify such outputs we propose to compare the aforementioned maps with the orthophotomaps produced from the UAV-acquired visual-light photographs taken in near real-time.

Although such a stepwise procedure is conceptually complete, there is no clear picture of whether it is possible to detect changes in water extent using the UAV-based orthophotomaps. Hence, this paper aims to check the meaningfulness of the UAV-based observations of water surface areas. In order to prove the aforementioned HFU concept we herein aim to verify the research hypothesis which reads as follows: "small changes in water surface areas are observable using the UAV". Such small changes may occur, for instance, when river stages rise from mean to high levels which does not always produce inundation (i.e. when water does not pass embankments or river banks, but only sinks into old river channels, flows through flood shortcuts or fills the current river channel). In order to explain such changes we graphically present the difference between water extents during low and high stages (Fig. 1). Since water surface area is directly associated with river stage (Usachev, 1983; Smith, 1997), our problem of detecting the above-mentioned changes is equivalent to seeking significant transitions in river stages. In other words, our hypothesis can also read as follows: "meaningful changes in river stages are observable using the UAV".

Both flood extents and water levels of large rivers are observable from satellites. For observing water surface area, the use is made of the following satellite-acquired measurements: high-resolution visual or infrared images, passive microwave data and radar images. For observing water levels from satellites researchers utilize: radar altimetry and high-resolution satellite imagery. Since 1997, when the extensive review of the above-mentioned methods was published (Smith, 1997), numerous studies on observing water surface areas and water levels using remote sensing techniques have been carried out (Cobby et al., 2001; Kouraev et al., 2004; Prigent et al., 2007; Schnebele et al., 2014). However, these approaches are targeted at large rivers, and there are few methods to observe water surface areas, and hence water levels, of small rivers. This paper aims to propose such an approach and to confirm its potential experimentally.

In order to verify the above-mentioned hypothesis we use a time series of five orthophotomaps produced from aerial photographs taken by the UAV at different hydrologic situations that occurred along the Ścinawka river (SW Poland). The observations were made in the experiment during which the same conditions of data acquisition were kept. We adopt a rigorous



statistical analysis, the use of which is essential in the process of detecting changes in water surface area. The remainder of this paper is organized as follows: the second section presents the study area and data, the subsequent section focuses on the UAV-based photo acquisition techniques and statistical methods, the fourth section shows the results and their discussion, while the last section concludes the paper.

## 2  Data

### 2.1  Study area

The research was conducted in the Southwestern Poland, in Kłodzko County located in the Central and the Eastern Sudetes (Fig. 2). The main river of the region is Nysa Kłodzka (left tributary of the Odra river), and one of its key tributaries is the Ścinawka river.

The channel system in Kłodzko County is directly related to the bedrock and refers to the tectonic structures and rock resistance. This region has unique topographical conditions – rivers and streams flow down from surrounding morphologically-differentiated mountain ranges to the Nysa Kłodzka Valley. The majority of the rivers in their upper sections are typical mountain streams. The middle and lower sections of the channels, located in the foothills of the mountains, present an alluvial character – wide channels with numerous large cutbanks, bars and meandering parts. The complex topography and the extensive hydrographic network contribute to rapid and catastrophic floods in this area (Dubicki et al., 2005; Kasprzak, 2010).

One of the alluvial-type river sections is a fragment of the Ścinawka river channel located directly upstream the Gorzuchów gauge (50°29′ N, 16°34′ E, 8+030 km of the course of the river). The headwaters of the Ścinawka river are located in the Wałbrzyskie Mountains (720 m a.s.l.), the river then flows to the Nysa Kłodzka river. The length of the Ścinawka river is approximately equal to 62 km. Its drainage basin has the area of 595 km$^2$. The valley is of Permian age, with a thick layer of the Pleistocene material superimposed (terraces are well visible in the bottom of the valley). The analyzed part of the channel has length of 1500 m and belongs to the lower part of the basin, within which the Ścinawka river flows through a wide valley (250–400 m), with a flat bottom. The river banks in the studied part are asymmetrical. The fragment of the channel is located between two gorge sections. Associated with the relatively more resistant rocks are sections where the valley is significantly narrowed. The width of the channel, in the investigated part, varies between 5 and 25 m, and the average slope of the channel is 3.4 per mil. This part of the river has winding character, and is contemporary formed mainly by lateral erosion. The analyzed section is located far from human settlements, hence regulation works have not been undertaken here. Soft material forming the river bed and banks as well as lack of engineering structures and occurrence of frequent flood episodes provide profound conditions for development of erosion and accumulation channel landforms. Evident channel bedforms are visible, for instance steep banks of erosional origins reveal heights and depths of 1.5–2 m. As regards accumulation-driven forms, various types of bars are common in the studied channel, with point bars and longitudinal ones that reflect the characteristics of a meandering stream.

The channel morphology of the studied part of the Ścinawka river may be linked to the pool-riffle channel pattern, with exposed bars and highly turbulent flow through riffles and more tranquil flow through pools. The average annual discharge at



the Gorzuchów gauge is approximately equal to 4.63 $\mathrm{m}^3$/s (1951–2011), whereas the average annual water level is of 71.6 cm (1981–2011).

In the studied fragment of the Ścinawka channel nine sites of variable lengths have been selected and coded as S1,S2,...,S9 (Fig. 3). Water extents in different seasons are analyzed in these sites. The observations have been carried out on: 27/11/2012, 13/05/2013, 21/08/2013, 27/09/2013 and 02/06/2014.

The selection of the sites is based on the following conditions: (1) all sites are covered by the observations from the above-mentioned five dates, (2) there are no distortions of the produced orthophotomaps (i.e. no data masks or incorrect peripheries, both arising from the limited coverage of aerial photos), (3) the sites are characterized by different morphological positions, cover channel forms, and enable to observe water extent, (4) the sites cannot be covered too lush vegetation which prevent the identification of the riverbank.

### 2.1.1 Site S1

The site is located in the central part of a large channel bend (length of the meander about 250 m, amplitude of 150 m, average radius about 90 m). The part of the channel is approximately 61 m long and its width varies between 13 and 15 m. Both banks are intensively eroded, especially the right one. Longitudinal underwater forms are visible. At the left bank, the exposed sand bar with vegetation is located. This large form is about 26 m long and 4 m wide. The channel banks are grown by bushes, grassy vegetation and individual trees.

### 2.1.2 Site S2

The site is also located in the central part of the aforementioned channel bend. The fragment is 56 m long and the channel width ranges from 12 to 19 m. The concave bank of the meander (right one) has been intensively eroded. The undercuts of 1.5 m height are visible. Longitudinal underwater sand bars have been formed, within which individual gravel elements and deposition shadows (downstream of obstruction bars) are visible. The channel banks are grown by bushes and grassy vegetation.

### 2.1.3 Site S3

The site covers the central part of the large meander. The fragment is approximately 80 m long and the width of the channel varies between 16 and 20 m. This part reveals a channel morphology which is characteristic for meandering rivers, with steep concave bank of erosional origin (right one) and convex bank, with a large point bar (about 11 m wide, 60 m long, with 32-metre part in the site) partially grown by grassy vegetation. The bank undercuts reach heights of approximately 2 m. The bank failures are also visible. Mid-channel underwater structures have been formed, with a large exposed one (45 m long and 9 m wide), partially grown. On their surface individual gravel elements with deposition shadows are visible. The left bank of the channel is grown by grassy vegetations and the right one by bushes and trees.



### 2.1.4 Site S4

The river flows straight through the fourth site. This part is about 50 m long and its width varies between 9 and 15 m. Alternately occurring channel narrowing and its extension are visible, with pool and riffle segments. The river banks are asymmetrical – right bank is higher and intensively eroded. Longitudinal sand bars are formed, with mid-channel and side ones. The banks are
grown by bushes and single trees. On the left end of the fragment, the beginning of an anabranch is visible.

### 2.1.5 Site S5

The river fragment is approximately 50 m long and is also located on a straight part of the channel. The width of the Ścinawka river varies here between 12 and 25 m. The banks are still asymmetrical. The right bank is intensively eroded – undercuts with height up to 1.5 m are visible, within which bank failure occurs (fragments of the bank are slid down to the base of the bank).
On the left side, the end of an anabranch is located. In this part of the channel, the left bank is more intensively eroded whereby the trees fall down to the river. In the second part of the site accumulation forms are common, i.e. there exists mid-channel, side, up- and downstream of obstruction bars. Both banks are grown by bushes and single trees.

### 2.1.6 Site S6

The short part of the channel (44 m long) is located at the beginning of the bend. The width of the channel changes between
10 and 18 m. Both banks are intensively eroded. On the left side of the channel deep cutbank (about 20 m long) exists, with ground slides to the base of the bank. Accumulation-driven forms are visible. There exist different types of bars, i.e. exposed mid-channel bars, side and mid-channel bars, up- and downstream of obstruction bars. The banks of the river are grown by grassy vegetation.

### 2.1.7 Site S7

The site covers a straight fragment between successive channel bends (the point of inflection), just upstream of the bridge. This fragment is approximately 55 m long and its width ranges from 16 to 21 m. The right bank is eroded, and on the left side of the channel sand-gravel side bar is formed (35 m long and 6 m width with its underwater parts). Longitudinal bedforms are located along the entire fragment. The banks of the channel are grown by grassy vegetation and single bushes.

### 2.1.8 Site S8

The eight site is also located between the above-mentioned channel bends. The studied part is approximately 31 m long and the width changes from 10 m to 24 m. The site is situated downstream of the bridge which forms a significant obstacle for free water flow. Downstream of obstruction bars are formed below the pillars of the bridge, with the sand-gravel bar (10 m long and about 4 m width) on the left side of the channel and organic deposition on every pillars. Downstream of the bridge wide riffle is formed. Both banks of the channel are grown by bushes and grassy vegetation.





### 2.1.9 Site S9

The site covers a straight part of the channel, downstream of the artificial rocky step, upstream the left-side channel bend. This part is about 52 m long and its the average width is equal to 13 m. Longitudinal sand and gravel bars are visible. Some of them are exposed. The banks of the channel are grown by bushes and grassy vegetation.

## 2.2 UAV data processing

We carried out the UAV survey using the swinglet CAM fixed-wing solution manufactured by senseFly. Swinglet CAM is lightweight (0.5 kg) and its payload includes a single consumer-grade RGB camera that records the photographs as JPG files. The individual pictures are geotagged. In order to produce orthophotomaps we process these files with the Structure-from-Motion (SfM) algorithm (Westoby et al., 2012) in the Photoscan software provided by AgiSoft, without use of ground control points (GCPs). We produced the georeferenced orthophotomaps in Photoscan which for the purpose of georeferencing uses coordinates extracted from the geotagged images. Such orthophotomaps were compared with the LIDAR digital terrain model (DTM), and we identified offsets between the two. The resolution of the LIDAR data was of 1 m, and the data acquisition was carried out in 2010. To remove the offsets we used the spline function in ArcMap 10.2.2 by ESRI. Use of LIDAR data to improve spatial references of UAV-acquired materials is known (Liu et al., 2007). However, these authors use LIDAR data to improve GSPs quality, and our approach is different as it applies splines to provide a spatial fix and correct for errors.

Although we did not use GCPs in the process of orthophotomap generation, we believe that for the purpose of the proposed analysis the absolute fit of orthophotomap to the Earth-fixed reference is not as crucial as the internal accuracy (within the orthophotomap). Indeed, the accuracy of area computation is not sensitive to linear motions of the cartographic source, but is vulnerable to the quality of the sources, and the latter is guaranteed by the above-mentioned LIDAR-based procedure that can be repeated at any time. Temporal analysis of landform topography is associated with a need of high accuracy which is guaranteed by the SfM-based materials (Clapuyt et al., 2015). We believe that our approach, carried out without use of GCPs, is also accurate because the accuracy is kept within every single orthophotomap as discussed above. We do not use the digital elevation model (DEM) of differences (DoD) and hence the highly accurate Earth-fixed reference is not needed.

Water extent during high flow, or inundation if overbank flow occurs, is identified on ortophotomaps as maximum range of terrain covered by water. Calculations of water surface areas were carried out to compare water extents recored on different dates. Thus, having a series of five orthophotomaps for each site (S1,S2,...,S9), corresponding to the above-mentioned dates of observations, we produced polygons in order to calculate the areas of water extent. To accurately carry out such a polygon generation procedure, the following method-related problems should be addressed: (1) the procedure to determine the edges of water extent should be well-documented to enable its repetition, (2) the accuracy of digitalization may vary across cartographic scales and experts, and hence its impact should be controlled, (3) there should be a recommended procedure to compute water surface area.

Morphology and vegetation were the most important factors in planning the procedure to determine the boundaries of water extent. The extent in question is relatively easy to identify if the water reaches clear barrier, such as for instance undercut river





banks, defense river banks, levees or dams. The water extent within plains as well as for rivers of indistinct and small slope banks, which additionally may be accompanied by bars, is difficult to determine. A similar problem refers to the banks of mid-channel forms.

Soil-turf overhangs on undercut river banks and clumps of grass growing on the low banks cover the line along which water meets the land (Fig. 4). Hence, the extent is determined as a line connecting the gaps between the clumps of grass (Fig. 5A and Fig. 5B). Bushes as well as small trees growing on the edge of a river channel also cover riverbank. Then, water extent is drawn as a line interpolated between last exposed parts of riverbank at both ends of woodlots or bushes (Fig. 5C and Fig. 5D). This procedure is verified against the field observations as well as against orthophotomaps for November 2012, January 2013 and December 2013, when the leaves fell down. Sections of river channel, for which it was impossible to determine the water extent in accordance with the procedure described above, were withdrawn from the analysis. The procedure enables the determination of water extent with certain approximation. Following the above discussion on the temporal analysis of landform topography without GCPs, it is assumed that determination errors are similar for all nine sites and five dates, which allows us to compare areas computed for different dates.

Digitalization of water extent was conducted using ArcMap 10.2.2. To accurately measure the area, the spatial data were transformed into the cylindrical equal area projection in secant normal aspect, with longitude of the central meridian at $13.5°$ W and standard parallels at $51°$ S and $51°$ N. Accuracy of digitalization (minimum dimension of digitized features, e.g. channels between clumps or width of islands) was 10 cm, while the resolution of ortophotomaps was approximately equal to 3 cm.

## 2.3 Riverflow data

As a hydrologic reference to the low-, normal- and high-flow situations we use the water level data recorded at the adjacent gauge in Gorzuchów (south of S8 in Fig. 3) which belongs to a larger Local System for Flood Monitoring, named LSOP (Lokalny System Osłony Przeciwpowodziowej). The observations at the gauge are carried out in the real-time fashion every 15 minutes.

Fig. 6 presents the hydrographs observed during one week when UAV flights were performed, with superimposed dots that highlight water level at the time of the UAV observation. In all cases the warning and emergency (alarm) water levels were depicted for reference.

In order to classify water levels into low-, normal-, and high-flow stages we analyzed the daily data from the same gauge, measured since the beginning of the records in 1981 by the Institute of Meteorology and Water Management – National Research Institute (IMGW-PIB). For the hydrologic years 1982–2014 (note that in Poland a hydrologic year begins on 1 November), hence for the period 01/11/1981–31/10/2014, we computed (in brackets abbreviations used in the Polish hydrologic nomenclature are given): (1) minimum from a time series of annual minimum water levels (NNW), (2) maximum from a time series of annual minimum water levels (WNW), (3) minimum from a time series of annual mean water levels (NSW), (4) maximum from a time series of annual mean water levels (WSW), (5) minimum from a time series of annual maximum water levels (NWW), (6) maximum from a time series of annual maximum water levels (WWW). Fig. 7 presents three graphs (annual min, annual mean, annual max) from which we extracted the above-mentioned characteristics. There exist three classes of water





levels – i.e. low, mean and high stages – and additionally a distinct intermediate class between mean and high stages (Fig. 7). In contrast, there is no intermediate class between low and mean stages since NSW is smaller than WNW. It is apparent form Figs. 6, 7 and Tab. 1 that we consider two low-stage situations (27/11/2012, 21/08/2013), one mean-stage situation (27/09/2013), one intermediate stage situation (02/06/2014) and one high-stage situation (13/05/2013).

## 5  3  Methods

Let us assume that we have $m$ UAV-based orthophotomaps that consist of observations of the same part of river channel carried out at times $t_1, \ldots, t_n$. Let us consider only such fragments of the orthophotomaps which meet the criteria outlined in Subsection 2.1. Thus, for each orthophotomap we obtain $n$ sites, coded as S1,...,Sn, in which water extent should be estimated. Such water-covered areas are expressed by polygons (coded as $s_1, \ldots, s_n$), the production of which should follow the procedure

outlined in Subsection 2.2. Having produced the polygons, we are able to calculate water surface areas. Hence, we consider a matrix $\mathbf{A}$:

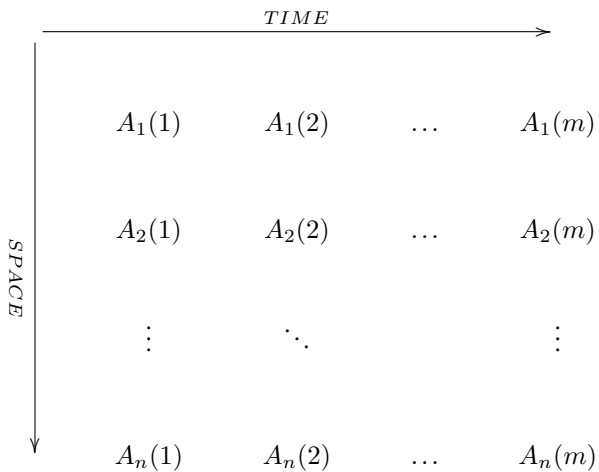

where $A_s(t)$ is water surface area in site $s$ at time $t$. In order to obtain the version of these data which is independent of sizes of S1,...Sn (and the corresponding polygons $s_1, \ldots, s_n$), we transform the areas by computing the ratio of water surface area at a

given section $s_i$ $(i = 1, \ldots, n)$ and at a given time $t_j$ $(j = 1, \ldots, m)$ to a sum of water surface areas in $m$ episodes that occurred at times $t_1, \ldots, t_m$ at the same site $s_i$. This ratio matrix, denoted by $\mathbf{R}$, is composed of the following elements:

$$R_i(j) = \frac{A_i(j)}{\sum_{k=1}^{m} A_i(k)}. \tag{1}$$

The transformation allows us to consider $m$ samples $R(1) = [R_1(1), \ldots, R_n(1)], \ldots, R(m) = [R_1(m), \ldots, R_n(m)]$, however, they should follow the i.i.d. (independent and identically distributed) property to be analyzed as statistical samples. The popular

transformation that helps to attain this goal is based on a logarithmic function which, in our case, produces a new matrix $\mathbf{L}$, the



elements of which are computed as $L_i(j) = \ln[R_i(j)]$. The corresponding samples are denoted as $L(1) = [L_1(1), \ldots, L_n(1)]$,...,
$L(m) = [L_1(m), \ldots, L_n(m)]$.

Let us now formulate a research hypothesis to be verified. Consider two samples $L(j)$ and $L(k)$, $j \neq k$ (but in practice $j < k$). We aim to check whether water surface areas at the time $k$ are significantly greater than at the time $j$. The hypothesis may be expressed in terms of means, i.e.

- H0: two samples have the same mean water surface areas,

- H1: the mean water surface area is greater in the subsequent, $L(k)$, sample than in the preceding one, $L(j)$.

Such a problem may be solved using the Student's t-test, however, numerous assumptions must be fulfilled prior to its application. Indeed, each of the samples $L(j)$ and $L(k)$ should be i.i.d., follow the normal distribution, and the two datasets should have the same variances. Several standard statistical tests can be used to check these assumptions:

- independence, Ljung-Box test (Ljung and Box, 1978),

- normality, Shapiro-Wilk test (Shapiro and Wilk, 1965; Royston, 1995),

- symmetry, D'Agostino test (D'Agostino, 1970),

- mesokurticity, Anscombe-Glynn test (Anscombe and Glynn, 1983),

- equality of variances, F test (Box, 1953).

If the assumptions are fulfilled, the Student's t-test can be applied to test H0 against H1. However, if the sample size is small what is often the case, the bootstrapped Student's t-test should be applied to verify the asymptotic results. Following the idea of Efron (1979), $B$ values of the Student's statistics should be computed, and their mean leads to the bootstrapped solution. If the bootstrapped solution confirms the asymptotic one, the decision on the hypothesis can be made.

## 4 Results and discussion

It is apparent from Fig. 8 that fragments of sites S1–S9 are covered with water, the extent of which is dissimilar at different dates of UAV observations. These dates correspond to low-, normal-, intermediate- and high-flow situations (see Subsection 2.3). Water surface areas in sites S1–S9 are juxtaposed in Tab. 2. Along the lines of Eqn. 1, Tab. 2 presents the ratios $R_i(j)$ and their logarithms $L_i(j)$ – for $i = 1, \ldots, 9$ and $j = 1, \ldots, 5$. The latter numbers become input data for the subsequent analysis.

Since we aim to compare five samples $L(1), \ldots, L(5)$, we first have to verify if they follow the i.i.d. structure. The p-values of the Ljung-Box, Shapiro-Wilk, D'Agostino and Anscombe-Glynn tests – juxtaposed in Tab. 3 – indicate that the five data sets are trajectories of statistical samples (sequences of i.i.d. random variables) which are normally distributed. This can be inferred at the significance level of 0.013 or smaller. In addition, variances between each pair of $L(1), \ldots, L(5)$ are shown to be similar





at the significance level of 0.03 or smaller, as the Fisher's test suggests (Tab. 4). The statistical inference unequivocally shows that the assumptions of the Student's t-tests are fulfilled.

Subsequently, we apply the Student's t-test to verify the above-mentioned hypothesis H0 against H1 (see Section 3), and we do so for each pair from $L(1), \ldots, L(5)$. The results are presented in Tab. 5 which juxtaposes p-values of the test, computed as asymptotic and bootstrapped solutions. Gray background boxes indicate statistically significant differences in water surface areas, which suggests the rejection of the H0 hypothesis. This means that the mean water surface area is shown to be greater in the subsequent $L(k)$ sample, $k = 2, \ldots, 5$, than in the preceding one.

It is known that water surface area is correlated with river stages. The characteristics of such relationships are reviewed by Smith (1997). Usually, the correlations are positive, with different degree of departure from a linear relation (Usachev, 1983). Hence, when water surface areas are analyzed in this paper in combination with river stages and their classes (Fig. 7) the following transitions are found to be meaningfully observable.

- Low stages (27/11/2012 and 21/08/2013) $\rightarrow$

    $\rightarrow$ mean stage (27/09/2013),

    $\rightarrow$ intermediate stage (02/06/2014),

    $\rightarrow$ high stage (13/05/2013).

- Mean stage (27/09/2013) $\rightarrow$

    $\rightarrow$ intermediate stage (02/06/2014),

    $\rightarrow$ high stage (13/05/2013).

- Intermediate stage (02/06/2014) $\rightarrow$

    $\rightarrow$ high stage (13/05/2013).

Noteworthy is the fact that the other transitions are found to be insignificant. To verify the adequacy of the detected changes between the water surface areas, we again refer to Fig. 6 and Tab. 1 which present stages observed at the Gorzuchów gauge at the time of the UAV observations. The visual examination of the graphs and table indicate that no changes in water-covered areas correspond to no changes in river stages and, conversely, significant differences in water surface areas observed by the UAV at dissimilar times correspond to visually well seen changes of water levels. This inference allows us to positively verify the research hypothesis stated in this paper. Namely, even small changes in water surface areas are observable using the UAV and – in addition – meaningful changes of river stages can also be extracted from the orthophotomaps based on the UAV-taken visual light photographs.

A note should be given on the accuracy of the elaborated approach. We believe that potential sources of error may reside in: (1) the SfM accuracy, (2) application of the SfM without GCPs, (3) problems with the determination of water boundaries due to presence of vegetation and undercuts. The quality of outputs from the SfM procedure depends on many factors – e.g. texture and light – and hence not uncommonly we produce incomplete orthophotomaps from well-overlapped photographs. Despite



these constraints the SfM procedure is well-established in scientific applications and is probably the most commonly used and accepted method for producing dense point clouds from the photographs taken by the consumer grade cameras (Westoby et al., 2012). As stated in Subsection 2.2, we do not use GCPs and believe that this does not undermine our approach. Indeed, we infer from multitemporal UAV-acquired data, similarly to Clapuyt et al. (2015) and Miřijovský and Langhammer (2015), and

calculate water surface areas which are accurate due to internal accuracy of every single orthophotomap (shifts of dissimilar orthophotomaps due to lack of GCPs are negligible). Similarly, Peter et al. (2014) omitted measuring GCPs in the field and identified them manually on external spatial data sources. Finally, a comment should be given on how vegetation and undercuts constrain the determination of water boundaries. In Subsection 2.2 we proposed a procedure to cope with the problem, but we are aware of its subjectivity. However, even satellite-based observations, using different sensors including radar, reveal similar

limitations (Smith, 1997).

## 5    Conclusions

We have shown that it is possible to detect even small changes in water surface area, using multitemporal orthophotomaps based on the UAV-acquired images. This can be done by the UAV equipped with the RGB consumer-grade camera which takes photographs with a sufficient overlap to produce the SfM-based dense point cloud and, consequently, an orthophotomap.

It is likely that transitions from normal or low flow to high flow does not produce large water surface area, as exemplified in Fig. 1. Our approach, verified in the experiment that uses the UAV data acquired at nine study sites during the campaign consisted of five observations, shows that it is possible to detect transitions between water surface areas produced by all characteristic water levels (low, mean, intermediate and high stages), and such a detection is statistically significant. Since water surface areas are correlated with river stages our approach can also be used as a tool for observing characteristic river

stages.

More specifically, we detected any rise of water level from low stages to: mean, intermediate and high stages. We also found any increase in water level from mean stages to: intermediate and high stages. Moreover, we detected rise of water level from intermediate stages to high stages. The detection was based on a rigorous statistical inference, based on several statistical tests performed in the asymptotic and bootstrap fashion.

Finally, it is important to identify potential applications of our approach. To do this we recall our motivation, stated in Section 1. We developed a solution, known as the HFU, which integrates the real-time flood warning system (HydroProg) with the hydrodynamical model (FloodMap) and their near real-time verification using UAVs (Niedzielski et al., 2014). A key problem in the HFU approach is the uncertainty of estimating water surface areas. The results presented in this paper unequivocally show that the UAV observes water extent with acceptable accuracy and, in addition, river stages can be inferred

from the observations. The later feature opens new perspectives for applications of the approach in the process of monitoring water levels of rivers in ungauged basins.





*Acknowledgements.* The research has been financed by the National Science Centre, Poland, through the grant no. 2011/01/D/ST10/04171 under leadership of Dr hab. Tomasz Niedzielski, Professor at the University of Wrocław, Poland. The UAV has been purchased from the statutory funds of the Institute of Geography and Regional Development of the University of Wrocław, Poland. The authors kindly acknowledge the authorities of the County Office in Kłodzko for productive partnership and making the data of the Local Flood Monitoring System
5 (Lokalny System Osłony Przeciwpowodziowej – LSOP) available for scientific purposes. We also acknowledge the Institute of Meteorology and Water Management – National Research Institute (IMGW-PIB) for providing us with access to hydrological data. We are also grateful to the authorities of the Commune Office in Kłodzko for access to the UAV take-off and landing site.



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





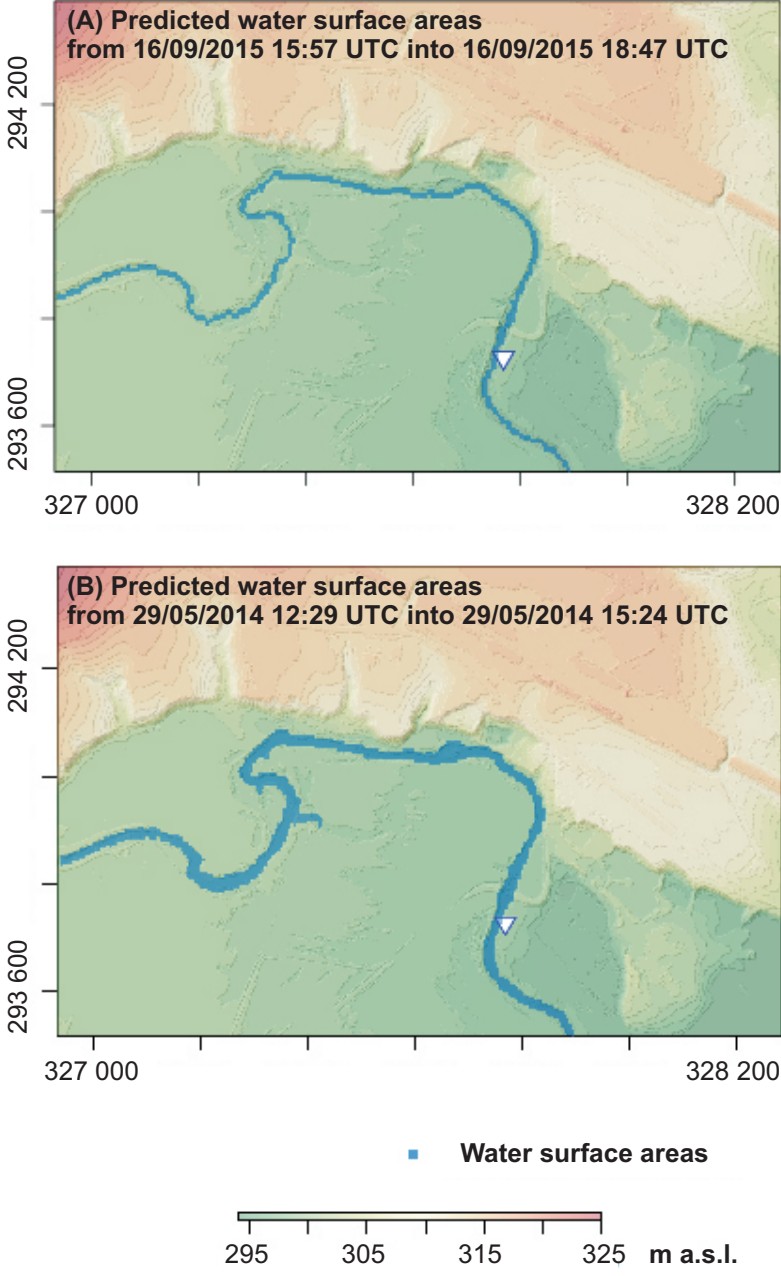

**Figure 1.** Predictions of water surface areas derived by the HydroProg-FloodMap solution extracted from the real-time web map service experimentally implemented for Kłodzko County: A – low flow situation, B – high flow situations.





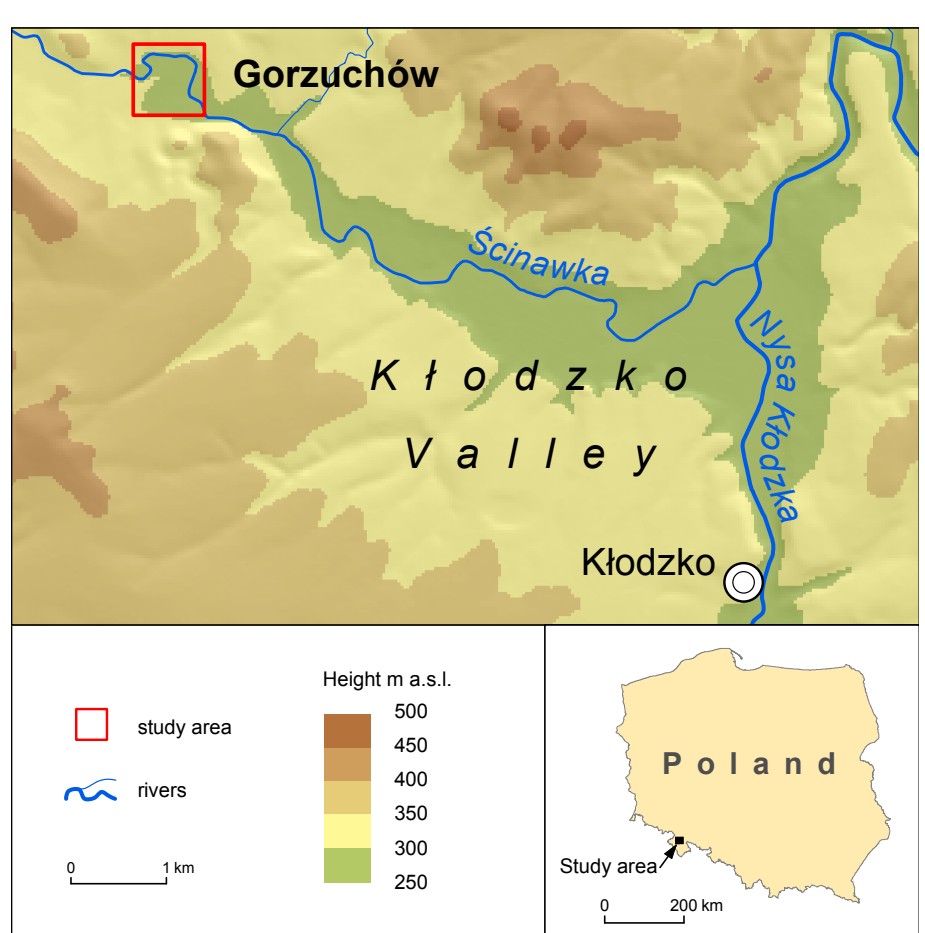

**Figure 2.** Map of study area.



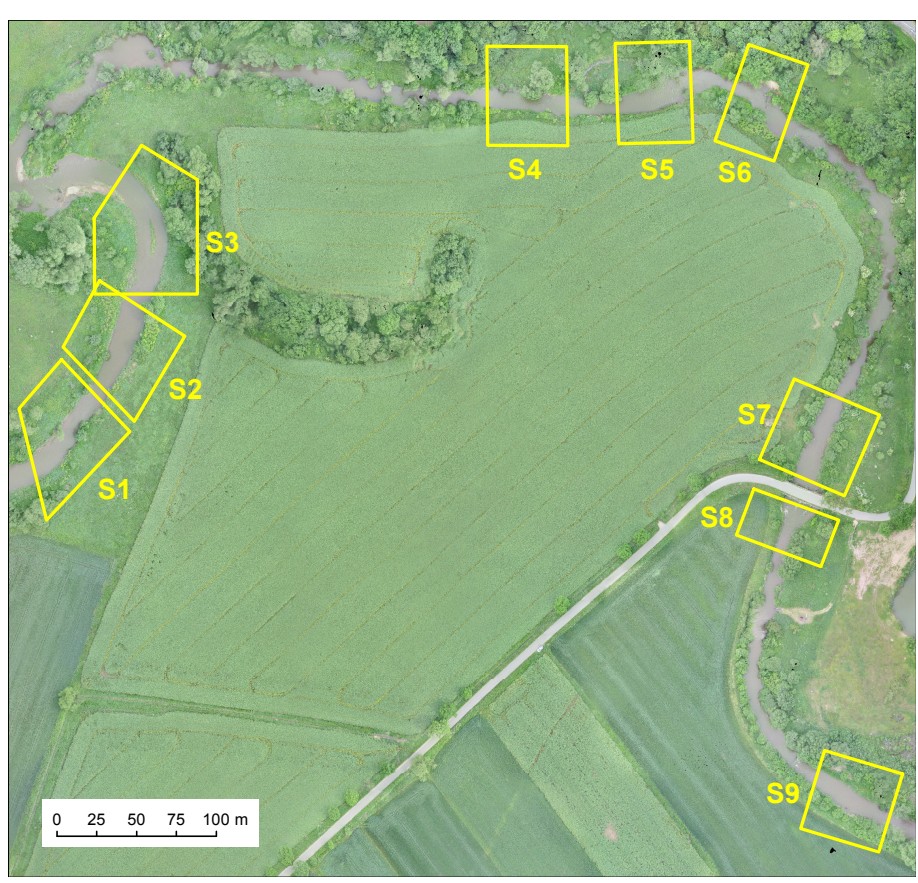

**Figure 3.** Study sites on the Ścinawka river in Gorzuchów. Numbers in site codes increase in a downstream direction.





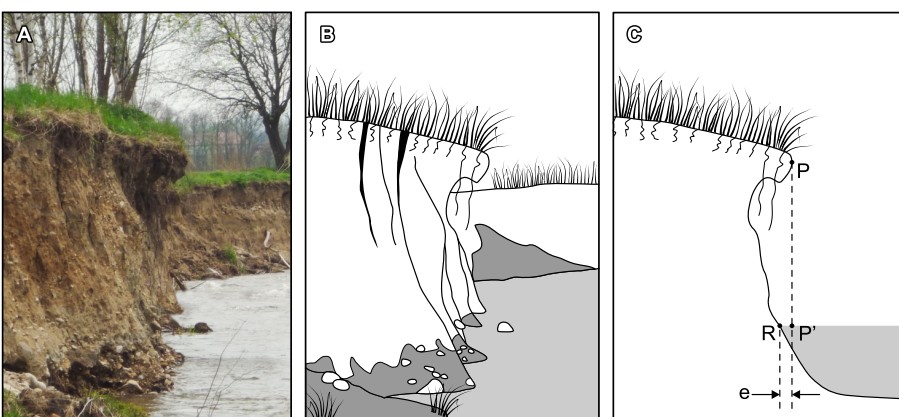

**Figure 4.** Undercut river bank: A – photograph, B – 3D sketch, C – 2D cross-section with edge P and its projection onto water surface P'
(water surface visible from the UAV) as well as true bank R (UAV-unobservable water surface, denoted on the figure as "e", stretches between
R and P').





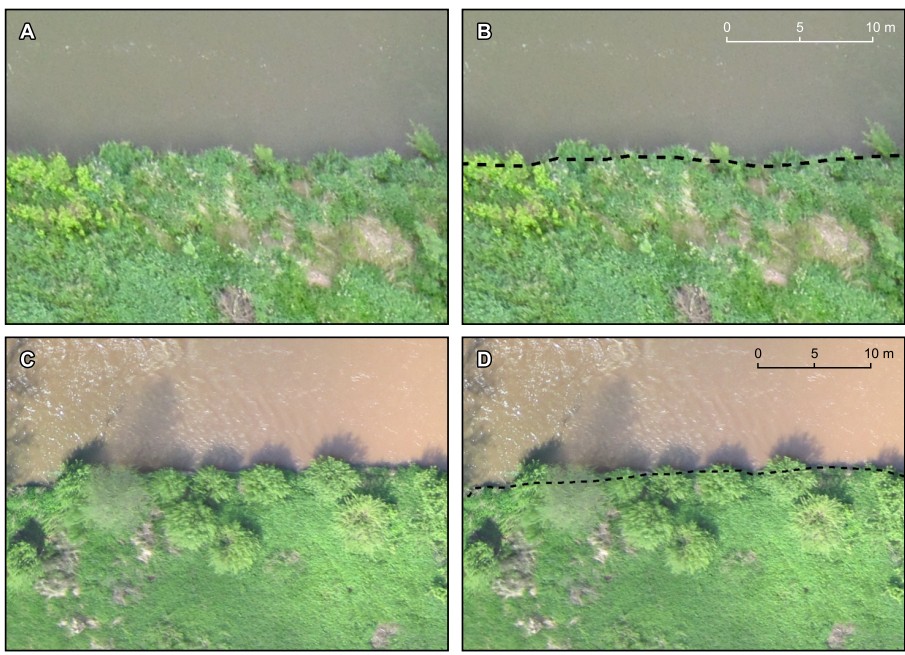

**Figure 5.** Determination of water surface areas: A and B – as a line connecting the gaps between the clumps of grass on undercut river bank covered by soil-turf overhangs, C and D – as a line interpolated between last exposed parts of riverbank at both ends of woodlots or bushes.





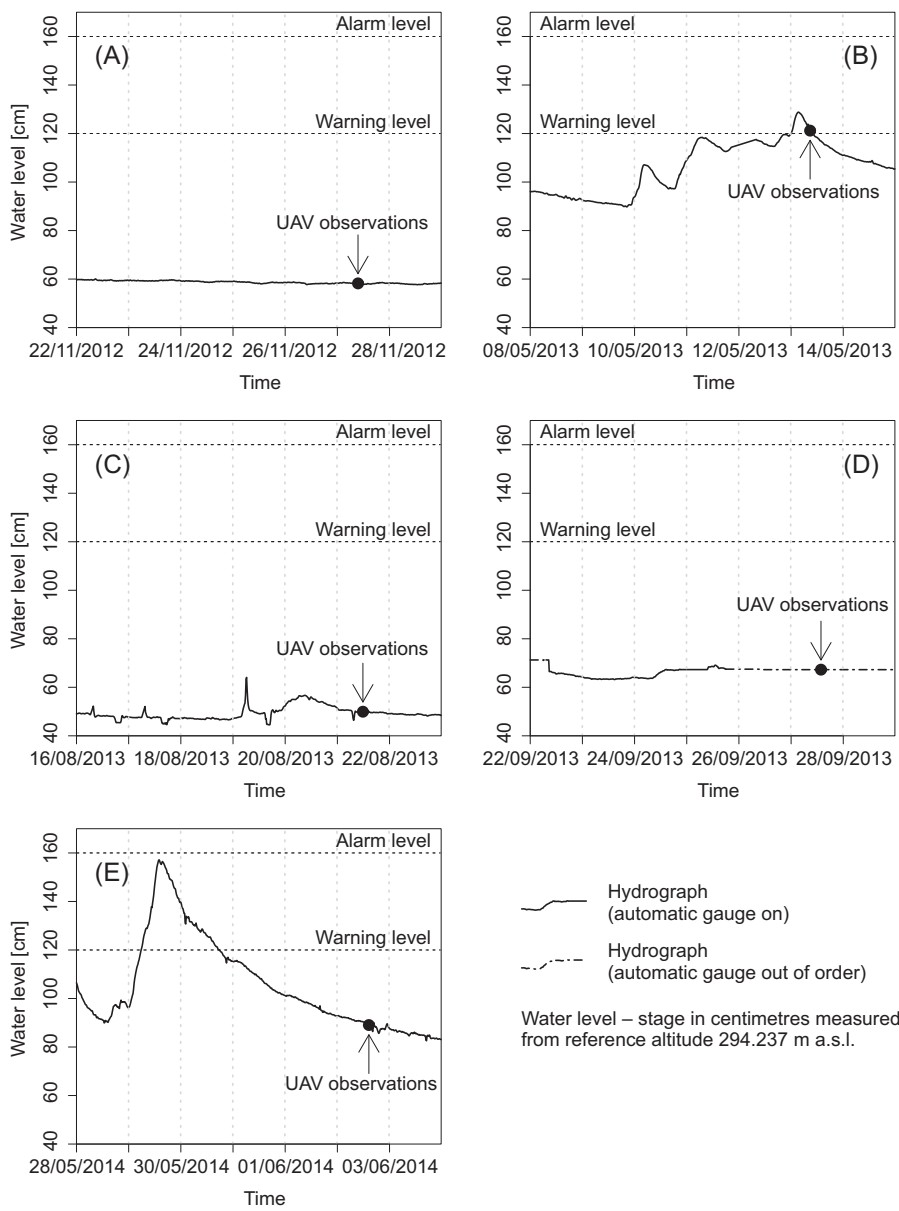

**Figure 6.** Hydrographs recorded in Gorzuchów before (approx. 5 days) and after (approx. 2 days) UAV flights on: A – 27/11/2012 , B – 13/05/2013, C – 21/08/2013, D – 27/09/2013, E – 02/06/2014.




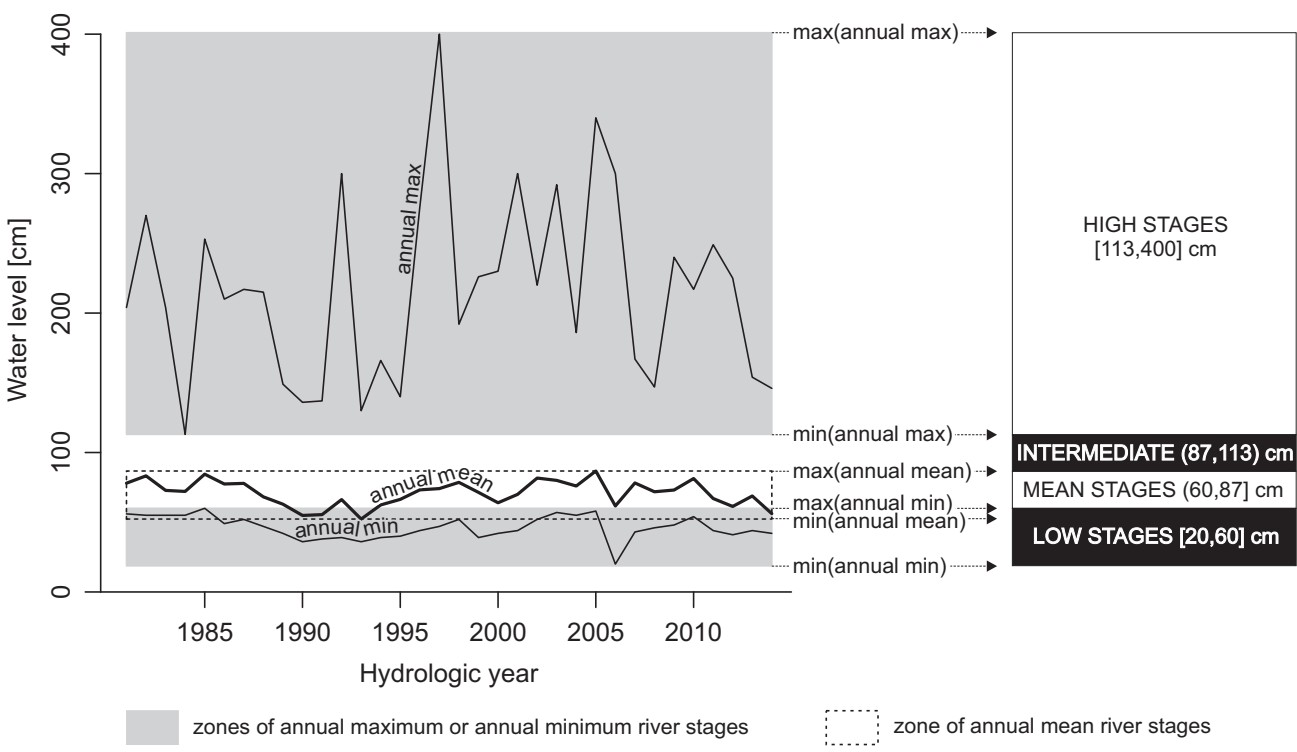

**Figure 7.** Time series of annual minimum, annual mean, annual maximum river stage computed for Gorzuchów in hydrologic years 1982–2014 (note that in Poland a hydrologic year begins on 1 November) along with their main statistics and the resulting river stage classes.





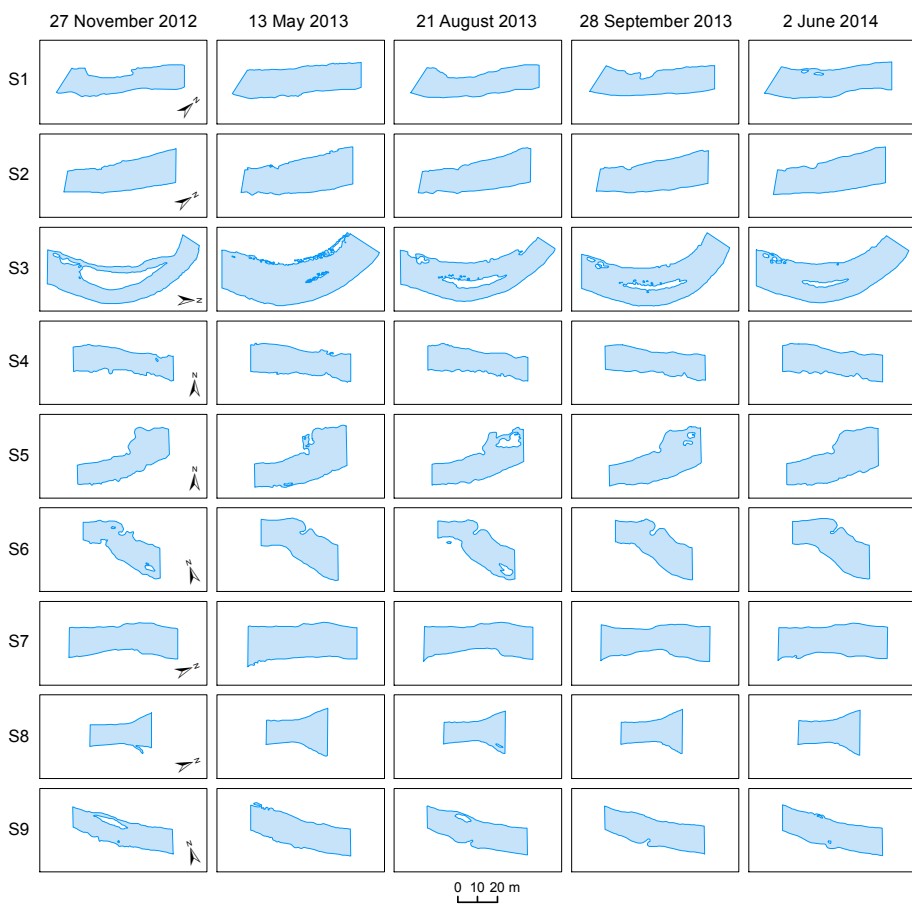

**Figure 8.** The water extents in the analyzed sections in the Ścinawka channel in Gorzuchów on: 27/11/2012, 13/05/2013, 21/08/2013, 27/09/2013 and 02/06/2014. Numbers of the sites (on the left) are the same as in Fig. 3.



**Table 1.** Water levels in Gorzuchów during the UAV flights on 27/11/2012, 13/05/2013, 21/08/2013, 27/09/2013 and 02/06/2014 along with description of hydrograph features and stage classification.

| Date | Water level in centimetres at UAV flight | Phase or shape of hydrograph at or around UAV flight | Stage classification based on Fig. 7 |
|---|---|---|---|
| 27/11/2012 | 58 | flat | low stage |
| 13/05/2013 | 121 | peak flow | high stage |
| 21/08/2013 | 50 | flat | low stage |
| 27/09/2013 | 67 | flat (uncertain) | mean stage |
| 02/06/2014 | 89 | recession limb | intermediate stage |



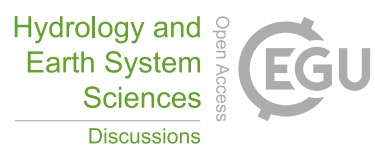

**Table 2.** Areas, fractions and logarithms, calculated according to Section 3, at dissimilar sites and dates of UAV flights.

| Site | 27/11/2012 | | | 13/05/2013 | | | 21/08/2013 | | | 27/09/2013 | | | 02/06/2014 | | |
|---|---|---|---|---|---|---|---|---|---|---|---|---|---|---|---|
| | Area | Fraction | Logarithm | Area | Fraction | Logarithm | Area | Fraction | Logarithm | Area | Fraction | Logarithm | Area | Fraction | Logarithm |
| 1 | 650.35 | 0.1740495 | -1.748415 | 835.01 | 0.2234691 | -1.498482 | 716.41 | 0.1917288 | -1.651673 | 721.52 | 0.1930964 | -1.644566 | 813.29 | 0.2176563 | -1.524838 |
| 2 | 790.08 | 0.1894899 | -1.663420 | 904.56 | 0.2169464 | -1.528105 | 794.73 | 0.1906051 | -1.657551 | 811.13 | 0.1945384 | -1.637125 | 869.01 | 0.2084202 | -1.568199 |
| 3 | 1025.64 | 0.1605611 | -1.829081 | 1559.12 | 0.2440759 | -1.410276 | 1193.84 | 0.1868923 | -1.677223 | 1288.73 | 0.2017471 | -1.600741 | 1320.52 | 0.2067237 | -1.576372 |
| 4 | 577.83 | 0.1873954 | -1.674534 | 664.41 | 0.2154741 | -1.534915 | 582.58 | 0.1889359 | -1.666348 | 602.13 | 0.1952761 | -1.633341 | 656.53 | 0.2129185 | -1.546846 |
| 5 | 651.93 | 0.1793238 | -1.718562 | 804.13 | 0.2211889 | -1.508738 | 628.35 | 0.1728378 | -1.755402 | 747.58 | 0.2056339 | -1.581658 | 803.50 | 0.2210156 | -1.509522 |
| 6 | 545.03 | 0.1820140 | -1.703672 | 654.68 | 0.2186319 | -1.520366 | 542.64 | 0.1812159 | -1.708066 | 602.50 | 0.2012062 | -1.603425 | 649.59 | 0.2169320 | -1.528171 |
| 7 | 778.58 | 0.1891130 | -1.665411 | 931.68 | 0.2263002 | -1.485893 | 779.56 | 0.1893510 | -1.664153 | 800.30 | 0.1943886 | -1.637896 | 826.89 | 0.2008472 | -1.605211 |
| 8 | 369.24 | 0.1807900 | -1.710419 | 456.38 | 0.2234561 | -1.498540 | 385.50 | 0.1887513 | -1.667325 | 403.04 | 0.1973394 | -1.622830 | 428.21 | 0.2096633 | -1.562252 |
| 9 | 584.76 | 0.1751838 | -1.741920 | 724.80 | 0.2171373 | -1.527225 | 658.02 | 0.1971312 | -1.623886 | 678.10 | 0.2031468 | -1.593826 | 692.30 | 0.2074009 | -1.573102 |

Area – water surface area in m$^2$

Fraction – water surface area as a fraction of total area of water-covered terrain at a given site (sum from all dates)

Logarithm – natural logarithm of fraction





**Table 3.** P-values of a few statistical tests applied to input data for the UAV flights on 27/11/2012, 13/05/2013, 21/08/2013, 27/09/2013 and 02/06/2014.

| Test | P-value for a given observation | | | | |
|---|---|---|---|---|---|
| | 27/11/2012 | 13/05/2013 | 21/08/2013 | 27/09/2013 | 02/06/2014 |
| Independence (Ljung-Box) | 0.059 | 0.092 | 0.444 | 0.713 | 0.828 |
| Normality (Shapiro-Wilk) | 0.208 | 0.013 | 0.178 | 0.321 | 0.863 |
| Symmetry (D'Agostino) | 0.265 | 0.076 | 0.247 | 0.758 | 0.979 |
| Mesokurticity (Anscombe-Glynn) | 0.193 | 0.017 | 0.132 | 0.171 | 0.759 |





**Table 4.** P-values of the Fisher's test applied to check if variances of input data are the same in each pair of the UAV observations.

| Date | P-value of Fisher's test between two observations | | | | |
|---|---|---|---|---|---|
| | 27/11/2012 | 13/05/2013 | 21/08/2013 | 27/09/2013 | 02/06/2014 |
| 27/11/2012 | 1.000 | 0.385 | 0.373 | 0.030 | 0.144 |
| 13/05/2013 | 0.385 | 1.000 | 0.980 | 0.170 | 0.536 |
| 21/05/2013 | 0.373 | 0.980 | 1.000 | 0.178 | 0.552 |
| 27/09/2013 | 0.030 | 0.170 | 0.178 | 1.000 | 0.440 |
| 02/06/2014 | 0.144 | 0.536 | 0.552 | 0.440 | 1.000 |



**Table 5.** P-values of asymptotic and bootstraped the Student's t-test applied to identify significant differences between water surface areas observed on 27/11/2012, 13/05/2013, 21/08/2013, 27/09/2013 and 02/06/2014. Grey boxes indicate statistically significant transitions.

| | 27/11/2012 | | 13/05/2013 | | 21/08/2013 | | 27/09/2013 | | 02/06/2014 | |
|---|---|---|---|---|---|---|---|---|---|---|
| | Asymptotic | Bootstrapped | Asymptotic | Bootstrapped | Asymptotic | Bootstrapped | Asymptotic | Bootstrapped | Asymptotic | Bootstrapped |
| 27/11/2012 | 0.5000000 | 0.5000000 | 0.0000000 | 0.0000000 | 0.0318682 | 0.0240133 | 0.0000377 | 0.0000100 | 0.0000002 | 0.0000000 |
| 13/05/2013 | 1.0000000 | 1.0000000 | 0.5000000 | 0.5000000 | 1.0000000 | 1.0000000 | 0.9999997 | 1.0000000 | 0.9978170 | 0.9987431 |
| 21/08/2013 | 0.9681318 | 0.9769138 | 0.0000000 | 0.0000000 | 0.5000000 | 0.5000000 | 0.0006099 | 0.0002194 | 0.0000007 | 0.0000001 |
| 27/09/2013 | 0.9999623 | 0.9999903 | 0.0000003 | 0.0000000 | 0.9993901 | 0.9997737 | 0.5000000 | 0.5000000 | 0.0000703 | 0.0000258 |
| 02/06/2014 | 0.9999998 | 1.0000000 | 0.0021830 | 0.0012121 | 0.9999993 | 0.9999999 | 0.9999270 | 0.9999738 | 0.5000000 | 0.5000000 |

Student's t-test with right-sided alternative; in Bootstrap $B = 10000$; gray boxes indicate statistically significant ($\alpha = 0.01$) changes in water surface areas.