# Peer review of "Observing river stages using unmanned aerial vehicles"

_Hydrology and Earth System Sciences, 2016_

## Referee Comment (RC1) · Anonymous Referee #1 · 22 Feb 2016

This paper is about mapping water surface areas reflecting different river stages using a simple UAV. The authors use the standard "Structure from Motion" method, however, without ground control points, and declare that the accuracy of mapping is sufficient to catch differences in the spatial extent of the river water levels due to a sufficiently high internal accuracy of the resulting orthophotomaps. The results indeed seem to be reasonable and supported by a statistical analysis, however, the paper needs to be more clearly written, some parts and methodological aspects are rather poorly explained.

Here are my suggestions:

- Please specify if all flight missions have been carried out using the same parameters, especially flight heights having the impact on the ortophotomap resolution and thus the accuracy of mapping

- Please explain in detail how you used the LIDAR data to do "a spatial fix and correct for errors". I think the statement that you have used the "spline function in ArcMap" is not sufficient. This part needs to be written more clearly what you have done here

- p. 2, l. 26, "high-resolution visual" is probably "high-resolution visible"

- on p. 6, l. 27-31 you have presented the criteria that should be met by the polygon generation procedure. As far as I have understood correctly, you have used a manual digitalization/vectorization of the water extent. Has this procedure met these criteria? Is the accuracy acceptable to catch relatively small variations in the extent of water surface?

- p. 6, l 15, explain "GSPs" abbreviation, probably a typo

- p. 9, l 3-7, please describe why you expect that water surface area at the time k will be greater than at the time j. Figure 6 clearly shows that water levels (A)-(E) are not increasing in time. This needs to be better explained because it is centred in the core of your paper with implications for your conclusions.

- on p. 10, l. 10-20 you have presented several transitions between stages. However, this part is very poorly explained. For example, how it is possible to have a subsequent stage (13/05/2013) after a later stage (02/06/2014).

- I recommend to present a Table with areas of the identified polygons representing identified water extents (equivalent to Figure 8). Some polygons seem to be almost identical and it is difficult to visually identify if it is larger or smaller.
* * *

---

## Referee Comment (RC2) · Anonymous Referee #2 · 27 Feb 2016

Firstly, this paper is at times difficult to understand due to the poor use of English language. I have made a few suggestions for improvement on the first page, however I think the text should be professionally edited before publication.

I think the use of aerial photos taken from UAVs to observe river stage could be a useful practice. I also think that this paper is not doing the idea sufficient justice. The aim/tested hypothesis is too basic to provide results of any impact. The tested hypothesis 'meaningful changes in river stages are observable using the UAV' is most likely going to be accepted as long as your data is of sufficiently high resolution and the channel morphology such that an increase in stage results in a larger water surface area. The first of these two requirements is solely dependent on the combination of UAV flying height and camera resolution, so can be adjusted until sufficient ground resolution has been achieved. The latter requirement is of course dependent on the channel

under observation. In most cases some increase will be visible with increasing stage and certainly in channels with less steep banks surface area will vary quite strongly. In any case this variability means that results found in this study are not necessarily transferable to other river systems with different channel cross sections.

The justification of the absence of GCPs is not fully clear to me. Regardless I think it is impossible to accurately draw polygons at the same location in images taken at different points in time, if these images cannot be georeferenced through a set of gcps visible in each image. You suggest each image is linked to a Lidar dataset, from which exact polygon locations may have been mapped on to the images, but again the explanation of this procedure is not clearly described. Line 28/29 p. 6 mentions that 'the procedure to determine the edges of water extent should be well-documented to enable its repetition', but there is no further mention of this procedure.

A further issue with the data is the independence of the observations. You did a test to test for the independence your samples, but I'm not sure this is appropriate (the test is in any case meant for time series not spatial dependence). Since the polygons were taken along the same river at relatively short distance, the measurements are to some extent going to be affected by spatial autocorrelation. The difference measured at the one location is going to be very similar to that measured at a nearby location. A t-test requires independence of observations, a requirement which is therefore not met.

As far as I'm aware it is not appropriate to do a series of t-tests on the data, as you increase your probability of making a Type I error. Instead an ANOVA style test should be done, with post-hoc tests to identify differences in means.

I think the study area description is too extensive, as a lot of the information relates to quite an extensive area while the study only focusses on a short section of a small stream in that area. Individual measurement site descriptions also take up a large part of the paper text, while the detail given is not overly relevant for the analysis.

Please also note the supplement to this comment:
http://www.hydrol-earth-syst-sci-discuss.net/hess-2016-49/hess-2016-49-RC2-supplement.pdf

**Supplement:**

[revised manuscript text omitted]

---

## Author Comment (AC1) · 25 Apr 2016

[REVIEWER #1] This paper is about mapping water surface areas reflecting different river stages using a simple UAV. The authors use the standard "Structure from Motion" method, however, without ground control points, and declare that the accuracy of mapping is sufficient to catch differences in the spatial extent of the river water levels due to a sufficiently high internal accuracy of the resulting orthophotomaps. The results indeed seem to be reasonable and supported by a statistical analysis, however, the paper needs to be more clearly written, some parts and methodological aspects are rather poorly explained.

[AUTHORS' RESPONSE] We thank the Reviewer #1 for assessing that the manuscript presents reasonable results. Having read the reviews offered by two Referees we entirely agree that the manuscript should be substantially modified so that some parts on methods are better explained. We hereby declare that, in our opinion, the improvement in question is doable, and the revised version of the text would reveal better quality and completeness.

[REVIEWER #1] Please specify if all flight missions have been carried out using the same parameters, especially flight heights having the impact on the orthophotomap resolution and thus the accuracy of mapping.

[AUTHORS' RESPONSE] Yes, indeed, the five flight missions have been performed with the comparable parameters. We double checked the UAV log files and confirmed that heights (both planned and measured during the mission), which determine the ground resolution, were kept approximately at a similar level. The values below present detailed vertical flight characteristics.

| A | B | C | D | E | F | G | H | I |
| --- | --- | --- | --- | --- | --- | --- | --- | --- |
| 27/11/2012 (1) | 109.0 | 296.4 | 113.6 | 109.2 | 410.0 | 405.6 | 451.0 | 447.5 |
| 27/11/2012 (2) | 109.0 | 296.5 | 112.4 | 109.1 | 408.9 | 405.7 | 453.5 | 447.9 |
| 13/05/2013 (1) | 109.0 | 297.7 | 116.0 | 108.9 | 413.7 | 406.6 | 456.2 | 449.9 |
| 13/05/2013 (2) | 109.0 | 299.8 | 118.1 | 109.0 | 417.9 | 408.8 | 458.6 | 450.4 |
| 21/08/2013 (1) | 109.0 | 301.1 | 117.0 | 108.2 | 418.2 | 409.3 | 458.8 | 450.6 |
| 21/08/2013 (2) | 109.0 | 295.2 | 115.0 | 108.9 | 410.2 | 404.1 | 450.0 | 444.8 |
| 27/09/2013 (1) | 109.0 | 294.5 | 114.8 | 108.8 | 409.3 | 403.4 | 452.7 | 455.5 |
| 27/09/2013 (2) | 109.0 | 295.5 | 114.5 | 109.1 | 410.0 | 404.6 | 454.3 | 446.4 |
| 02/06/2014 (1) | 109.0 | 305.3 | 115.3 | 108.3 | 420.7 | 413.6 | 452.7 | 461.0 |
| 02/06/2014 (2) | 109.0 | 294.3 | 114.3 | 108.9 | 408.6 | 403.1 | 451.1 | 445.4 |

A – Date and number of flight

B – Planned height above takeoff location [m]

C – Takeoff altitude [m a.s.l.]

D – Maximum height [m]

E – Mean height [m]

F – Maximum altitude [m a.s.l.]

G – Mean altitude [m a.s.l.]

H – Maximum altitude WGS84 [m]

I – Mean altitude WGS84 [m]

Since the information on stability of height parameters is very important for a complete understanding of comparability between the consecutive UAV missions, we produced a new table (expended version of the above-mentioned table) that might be included into the revised version of the manuscript.

[REVIEWER #1] Please explain in detail how you used the LIDAR data to do "a spatial fix and correct for errors". I think the statement that you have used the "spline function in ArcMap" is not sufficient. This part needs to be written more clearly what you have done here.

[AUTHORS' RESPONSE] In order to response to this comment we firstly put an emphasis on our key assumption which may be formulated as follows: "presence of a potential shift between two spatial data sets does not cause meaningful changes in area of the considered objects" (this is expressed in line 5 on page 11 of our HESS Discussion Paper). For instance, if one replicates an orthophotomap and applies a translation vector to such a newly produced spatial data set, the same objects will reveal the same areas (no change in scale and rotation). To support this finding we

refer to a recent paper by Mesas-Carrascosa et al. (2014) [Mesas-Carrascosa F.J., Notario-García M.D., Meroño de Larriva J.E., Sánchez de la Orden M., García-Ferrer A., 2014. Validation of measurements of land plot area using UAV imagery. International Journal of Applied Earth Observation and Geoinformation 33, 270–279]. These authors argue that "[. . .] Other shortcomings include the lack of vertical adjustment of the aerial camera and the unknown or variable interior orientation of the camera. These factors affect point position accuracy but do not necessarily decrease the accuracy of area measurements.[. . .]". Having justified a stability of area measurements in case of smaller point position accuracy, i.e. also in case of shift of orthophotomaps produced without ground control points (GCPs), we hereby describe the spline-based procedure that fixes all orthophotomaps to a single LIDAR data.

We identified characteristic features in the LIDAR digital terrain model (DTM) which were evenly distributed and possible to identify in the orthophotomap. These features comprise: crossings of bounds, crossings of drainage ditches, and centres of bridges or passages (crossings of streams and roads). More than 10 points were used to perform georeferencing, as the spline method requires. A spline function allowed us to precisely georeference the control points (i.e. the aforementioned mutual features) and transform raster data set with continuity and smoothness, such as the rubber sheeting method.

[REVIEWER #1] P. 2, l. 26, "high-resolution visual" is probably "high-resolution visible".

[AUTHORS' RESPONSE] We agree that the sentence does not read well in the initial version of the manuscript. We propose the following formulation of the sentence: "[. . .] For observing water surface area, the use is made of the following satellite-acquired measurements: HIGH-RESOLUTION VISIBLE LIGHT IMAGES OR INFRARED IMAGES, passive microwave data and radar images.[. . .]". We went through several research papers and double checked that the notion of "visible light images" is properly used in the above revised proposition.

[Figure]

[REVIEWER #1] On p. 6, l. 27-31 you have presented the criteria that should be met by the polygon generation procedure. As far as I have understood correctly, you have used a manual digitalization/vectorization of the water extent. Has this procedure met these criteria? Is the accuracy acceptable to catch relatively small variations in the extent of water surface?

[AUTHORS' RESPONSE] The Reviewer #1 pointed out an important problem of the accuracy of a manual vectorization carried out under several conditions (lines 27-31 on page 6 and the subsequent part of Subsection 2.2 in our HESS Discussion Paper). One of the most important factors that may potentially constrain a vectorization accuracy is related to vegetation. Mapping vegetation with UAVs becomes popular as a recent paper by Husson et al. (2014) shows [Husson E., Hagner O., Ecke F., 2014. Unmanned aircraft systems help to map aquatic vegetation. Applied Vegetation Science 17, 567–577]. These authors focus on delineating edges between water and non-submerged aquatic as well as riparian species. They write that "[...] In practise delineation was done by hand on paper printouts [...]" and "[...] Vegetation mapping, i.e. digitizing the UAS orthoimages, was performed manually by a human interpreter in a GIS using ArcGIS software [...]". Although we concentrate on a fluvial environment, the idea behind our manual expert-based vectorization remains similar to what Husson et al. (2014) propose. It is worth noting that our vectorization was practically carried out by two experts (GIS specialist + fluvial geomorphologist). Given this introduction, we unequivocally reply that the procedure met the assumed criteria (this is attained by the expert-based vectorization). We also believe that the accuracy of the produced water surface area is acceptable. However, it was our intention to include Fig. 4 and Fig. 5 which help the reader to identify potential sources of errors.

[REVIEWER #1] P. 6, l 15, explain "GSPs" abbreviation, probably a typo.

[AUTHORS' RESPONSE] Yes, indeed, the "GSPs" is a typo and in the revised manuscript should be replaced by "GCPs".

[REVIEWER #1] P. 9, l 3-7, please describe why you expect that water surface area at the time k will be greater than at the time j. Figure 6 clearly shows that water levels (A)-(E) are not increasing in time. This needs to be better explained because it is centred in the core of your paper with implications for your conclusions.

[AUTHORS' RESPONSE] Being grateful for notifying that the formulation of the hypotheses H0 and H1 is unclear, we hereby clarify the issue and propose how to make the part more comprehensive in the revised manuscript. The two sample Student's t-test is used to test a null hypothesis (H0) that means of two samples are equal, but three alternative hypotheses (H1) are allowed. These three alternatives include: means of two samples are different, mean of the FIRST sample is bigger than mean of the SECOND sample, mean of the SECOND sample is bigger than mean of the FIRST sample. In the latter two alternatives, the order of samples is important and has impact on where rejection region is located. Knowing the aforementioned basics, we stated the research hypothesis H0 with its alternative H1 on purpose, in the way that rejection of the null hypothesis implies acceptance of the alternative one (and this unequivocally indicates which area is meaningfully bigger). To clarify the entire problem, we suggest to consider two combinations of L(j) and L(k) (notations are explained in our HESS Discussion Paper). Recall that we test if mean[L(j)] = mean[L(k)] with the alternative that mean[L(j)] < mean[L(k)].

CASE 1

j = '27/11/2012'

k = '13/05/2013'

mean[L('27/11/2012')] = −1.71727

mean[L('13/05/2013')] = −1.501393

Arithmetically, mean[L('27/11/2012')] IS SMALLER THAN mean[L('13/05/2013')]. This inequality has also been confirmed statistically (the Student's t-test) at the significant

level of 0.01 (see Tab. 5, grey box indicates that the difference in means is statistically significant). Hence, in this case a subsequent episode (time step k) revealed meaningfully bigger water surface area than the preceding one (time step j).

CASE 2

j = '13/05/2013'

k = '27/11/2012'

mean[L('27/11/2012')] = −1.71727

mean[L('13/05/2013')] = −1.501393

Arithmetically, mean[L('13/05/2013')] IS NOT SMALLER THAN mean[L('27/11/2012')]. If we test (mean[L(j)] = mean[L(k)] with alternative mean[L(j)] < mean[L(k)]) we cannot reject the null hypothesis with the Student's t-test at the significance level of 0.01. Hence in this case a subsequent episode (time step k) does not reveal a meaningfully bigger water surface area than the preceding one (time step j).

Since Tab. 5 juxtaposes all cases of the above type (j does not equal to k), we suggest to remove the following phrase "[...] (but in practice j < k) [...]" from lines 3–4 on page 9 of our manuscript. The deletion of the sentence will make the conclusions straightforward.

[REVIEWER #1] On p. 10, l. 10-20 you have presented several transitions between stages. However, this part is very poorly explained. For example, how it is possible to have a subsequent stage (13/05/2013) after a later stage (02/06/2014).

[AUTHORS' RESPONSE] As we explained above, the order of L(j) and L(k) matters and influences the final results, however it is not necessary that j < k. The stages and transitions listed on page 10 are examples of low, mean, intermediate and high water levels. They have been recorded by real UAV flights on different dates. We used the UAV-observed water surface areas as true data that represent the aforementioned

stages. We believe that, for the analysis that aims to check the procedure proposed in this paper, the chronological order of transitions between stages is not important. Of course, we agree with the Reviewer #1 that it would be ideal to have the chronological set of transitions, however such a data set is not available. Thus, we mixed the order to check various POTENTIAL combinations of transitions. We think that the revised version of the manuscript should explain in a new paragraph which we are ready to prepare.

[REVIEWER #1] I recommend to present a Table with areas of the identified polygons representing identified water extents (equivalent to Figure 8). Some polygons seem to be almost identical and it is difficult to visually identify if it is larger or smaller.

[AUTHORS' RESPONSE] The areas, fractions and logarithms (hence all input data used for the analysis) have already been juxtaposed in Tab. 5 of the initial version of the manuscript.

---

## Author Comment (AC2) · 25 Apr 2016

[REVIEWER #2] Firstly, this paper is at times difficult to understand due to the poor use of English language. I have made a few suggestions for improvement on the first page, however I think the text should be professionally edited before publication.

[AUTHORS' RESPONSE] We accept the criticism and thank the Reviewer #2 for spotting linguistic problems. We are ready to correct the text according to the marked suggestions and edit the manuscript before publication.

[REVIEWER #2] I think the use of aerial photos taken from UAVs to observe river stage could be a useful practice. I also think that this paper is not doing the idea sufficient justice. The aim/tested hypothesis is too basic to provide results of any impact. The tested hypothesis 'meaningful changes in river stages are observable using the UAV'

is most likely going to be accepted as long as your data is of sufficiently high resolution and the channel morphology such that an increase in stage results in a larger water surface area. The first of these two requirements is solely dependent on the combination of UAV flying height and camera resolution, so can be adjusted until sufficient ground resolution has been achieved. The latter requirement is of course dependent on the channel under observation. In most cases some increase will be visible with increasing stage and certainly in channels with less steep banks surface area will vary quite strongly. In any case this variability means that results found in this study are not necessarily transferable to other river systems with different channel cross sections.

[AUTHORS' RESPONSE] We thank the Reviewer #2 for finding our concept of reconstructing river stages using UAV-taken photographs a useful practice. We accept the criticism, but we believe that – after implementing the suggestions kindly offered by two Referees – the revised manuscript is likely to do the above-mentioned idea sufficient justice. We hereby response to two key comments (height/resolution issues and channel properties) in the itemized points below.

– Indeed, flight altitude (and the resulting ground resolution) influences the water surface area observations. Our flight characteristics were kept uniform over the five observational campaigns, hence the resolution is also stable over the exercises. The similar comment has been also offered by the Reviewer #1. We double checked the flight logs and confirmed the comparability of UAV height parameters. The statistics calculated from the log data are juxtaposed in the following table (also included in the response to comments raised by the Reviewer #1).

| A | B | C | D | E | F | G | H | I |
|---|---|---|---|---|---|---|---|---|
| 27/11/2012 (1) | 109.0 | 296.4 | 113.6 | 109.2 | 410.0 | 405.6 | 451.0 | 447.5 |
| 27/11/2012 (2) | 109.0 | 296.5 | 112.4 | 109.1 | 408.9 | 405.7 | 453.5 | 447.9 |
| 13/05/2013 (1) | 109.0 | 297.7 | 116.0 | 108.9 | 413.7 | 406.6 | 456.2 | 449.9 |
| 13/05/2013 (2) | 109.0 | 299.8 | 118.1 | 109.0 | 417.9 | 408.8 | 458.6 | 450.4 |
| 21/08/2013 (1) | 109.0 | 301.1 | 117.0 | 108.2 | 418.2 | 409.3 | 458.8 | 450.6 |
| 21/08/2013 (2) | 109.0 | 295.2 | 115.0 | 108.9 | 410.2 | 404.1 | 450.0 | 444.8 |
| 27/09/2013 (1) | 109.0 | 294.5 | 114.8 | 108.8 | 409.3 | 403.4 | 452.7 | 455.5 |
| 27/09/2013 (2) | 109.0 | 295.5 | 114.5 | 109.1 | 410.0 | 404.6 | 454.3 | 446.4 |
| 02/06/2014 (1) | 109.0 | 305.3 | 115.3 | 108.3 | 420.7 | 413.6 | 452.7 | 461.0 |
| 02/06/2014 (2) | 109.0 | 294.3 | 114.3 | 108.9 | 408.6 | 403.1 | 451.1 | 445.4 |

A – Date and number of flight

B – Planned height above takeoff location [m]

C – Takeoff altitude [m a.s.l.]

D – Maximum height [m]

E – Mean height [m]

F – Maximum altitude [m a.s.l.]

G – Mean altitude [m a.s.l.]

H – Maximum altitude WGS84 [m]

I – Mean altitude WGS84 [m]

Since an extensive data set of UAV flight data is available, we are ready to include a new table that juxtaposes the characteristics in question. We believe that – given the fact that altitudes (and hence resolutions) were kept comparable – the properties of flight parameters did not undermine our inference.

– We are grateful the Reviewer #2 for mentioning the influence of channel morphology, mainly the slope of banks, on the observation of water surface area with the UAV. We

agree that such a relationship exists. We also entirely accept the comment that the results, prepared for a specific river in the SW Poland, are not transferable to other rivers with different cross-sectional parameters. In fact this has been already pointed out in the context of the relationship between water surface areas and river stages by Smith (1997) who argues that "[. . .] Until additional empirical rating curves relating inundation area to ground measurements of stage or discharge are made, it is difficult to assess their potential for extrapolation to other rivers of similar morphology. However, it seems likely that such curves will vary significantly between rivers and therefore must be constructed for each site. [. . .]" [Smith L.C, 1997. Satellite remote sensing of river inundation area, stage, and discharge: a review. Hydrological Processes 11, 1427–1439]. However, our approach is centred on a statistical analysis of water surface areas, not river stages themselves. In fact, we quantitatively infer on statistically meaningful changes in water surface area (this is a key part of our procedure) and only qualitatively, through the existence of a relationship between water surface areas and river stages published by Usachev (1983) [Usachev V.F., 1983. Evaluation of food plain inundations by remote sensing methods. In: Proceedings of the Hamburg Symposium, IAHS Publ. 145, 475–482], extrapolate our results into changes in river stages. We believe that our quantitative approach (recall that this concerns seeking changes in water surface areas in the orthophotomaps produced from the UAV-taken photographs) forms a general method that – under several conditions clearly identified in the manuscript – may be applied in other regions. However, the use of the approach to infer on river stages should be made with caution, since such an extrapolation requires a knowledge about the relationship between water surface areas and river stages (and the characteristics of this relation are vulnerable to sites-specific river morphology, especially bank slopes). We think that such constraints should be included in the revised manuscript, in particular in the discussion or conclusions.

[REVIEWER #2] The justification of the absence of GCPs is not fully clear to me. Regardless I think it is impossible to accurately draw polygons at the same location in images taken at different points in time, if these images cannot be georeferenced

through a set of gcps visible in each image. You suggest each image is linked to a Lidar dataset, from which exact polygon locations may have been mapped on to the images, but again the explanation of this procedure is not clearly described. Line 28/29 p. 6 mentions that 'the procedure to determine the edges of water extent should be well-documented to enable its repetition', but there is no further mention of this procedure.

[AUTHORS' RESPONSE] Yes, we are aware of possible problems that may be associated with measuring the area of polygons that are generated on a basis of orthophotomaps produced without GCPs. We also know that a procedure for delineating boundaries of water surface area needs to be clarified. The similar remarks have been also offered by the Reviewer #1, and the responses below use the arguments which we raised when replying to the comments provided by the Referee #1. In the two itemized points, we focus on: (1) area computation when GCPs are unavailable, (2) delineating edges of water surface areas.

– In line 5 on page 11 of our HESS Discussion Paper we argue that a presence of a potential shift between two spatial data sets does not cause meaningful changes in area of the considered objects. The arguments that support this statement can be found in a recent paper by Mesas-Carrascosa et al. (2014) [Mesas-Carrascosa F.J., Notario-García M.D., Meroño de Larriva J.E., Sánchez de la Orden M., García-Ferrer A., 2014. Validation of measurements of land plot area using UAV imagery. International Journal of Applied Earth Observation and Geoinformation 33, 270–279]. These authors argue that "[. . .] Other shortcomings include the lack of vertical adjustment of the aerial camera and the unknown or variable interior orientation of the camera. These factors affect point position accuracy but do not necessarily decrease the accuracy of area measurements. [. . .]". Having justified a stability of area measurements in case of smaller point position accuracy, i.e. also in case of shift of orthophotomaps produced without GCPs, we hereby describe the spline-based procedure that fixes all orthophotomaps to a single LIDAR data. We identified characteristic features in the LIDAR DTM which

were evenly distributed and possible to identify in the orthophotomap. These features comprise: crossings of bounds, crossings of drainage ditches, and centres of bridges or passages (crossings of streams and roads). More than 10 points were used to perform georeferencing, as the spline method requires. A spline function allowed us to precisely georeference the control points (i.e. the aforementioned mutual features) and transform raster dataset with continuity and smoothness, such as the rubber sheeting method.

– In lines 27-31 on page 6 and the subsequent part of Subsection 2.2 in the HESS Discussion Paper we listed three criteria labelled as (1), (2) and (3). While the latter two issues are associated with GIS methods (the same scale should be kept when carrying out a vectorization procedure and a cartographic projection as well as reference system should be unified before measuring areas), the first one is strongly related to environmental factors, mainly vegetation. This first statement has been explicitly highlighted by the Reviewer #2 as an element that needs to be clarified. Mapping vegetation with UAVs becomes popular as a recent paper by Husson et al. (2014) shows [Husson E., Hagner O., Ecke F., 2014. Unmanned aircraft systems help to map aquatic vegetation. Applied Vegetation Science 17, 567–577]. These authors focus on delineating edges between water and non-submerged aquatic as well as riparian species. They write that "[. . .] In practise delineation was done by hand on paper printouts [. . .]" and "Vegetation mapping, i.e. digitizing the UAS orthoimages, was performed manually by a human interpreter in a GIS using ArcGIS software". Although we concentrate on a fluvial environment, the idea behind our manual expert-based vectorization remains similar to what Husson et al. (2014) propose. It is worth noting that our vectorization was practically carried out by two experts (GIS specialist + fluvial geomorphologist). Given this introduction, we unequivocally reply that the procedure met the assumed criteria (this is attained by the expert-based vectorization). We also believe that the accuracy of the produced water surface area is acceptable. However, it was our intention to include Fig. 4 and Fig. 5 which help the reader to identify potential sources of errors.

[REVIEWER #2] A further issue with the data is the independence of the observations. You did a test to test for the independence your samples, but I'm not sure this is appropriate (the test is in any case meant for time series not spatial dependence). Since the polygons were taken along the same river at relatively short distance, the measurements are to some extent going to be affected by spatial autocorrelation. The difference measured at the one location is going to be very similar to that measured at a nearby location. A t-test requires independence of observations, a requirement which is therefore not met.

[AUTHORS' RESPONSE] All assumptions of the Student's t-test have been checked using: the Ljung-Box test (independence), the Shapiro-Wilk test (normality), the D'Agostino test (symmetry as a feature of the Gaussian distribution), the Anscombe-Glynn test (mesokurticity as a feature of the Gaussian distribution). The tests, performed with the significance level of 0.01, suggest that each sample (we analyze 5 samples corresponding to five dates) is "internally" independent is normally distributed. In particular, the Ljung-Box test provides arguments for independence since p-values are equal to 0.059, 0.092, 0.444, 0.713, 0.828 (Tab. 3 in our HESS Discussion Paper), for five consecutive dates. Hence, from a definition of statistical independency we infer that they cannot reveal autocorrelation. We would like to take this opportunity and put an emphasis on spatial independence which has not been investigated in our work. In addition, variances of paired data sets have been found to be similar (Tab. 4 in the HESS Discussion Paper).

[REVIEWER #2] As far as I'm aware it is not appropriate to do a series of t-tests on the data, as you increase your probability of making a Type I error. Instead an ANOVA style test should be done, with post-hoc tests to identify differences in means.

[AUTHORS' RESPONSE] Indeed, the ANOVA test is a generalization of the t-test to more than two groups. However, although we process five samples our intention is to allow pairwise comparisons. In other words, our approach is targeted at solving a simple operational problem: we have two sets of UAV-acquired observations carried
out on two different dates, and would like to know if water surface area increased in comparison to the preceding observation. In our manuscript we simply have 5 observations which allows us to carry out many tests to make the inference more evident. However, pairwise comparison is a fundamental feature of our procedure.

[REVIEWER #2] I think the study area description is too extensive, as a lot of the information relates to quite an extensive area while the study only focusses on a short section of a small stream in that area. Individual measurement site descriptions also take up a large part of the paper text, while the detail given is not overly relevant for the analysis.

[AUTHORS' RESPONSE] We entirely agree with the Reviewer #2 that the description of the study area is too detailed. In the revised manuscript, the information on study area will be significantly shortened and condensed so that the reader focuses on the method and its application.

[REVIEWER #2] Please also note the supplement to this comment: http://www.hydrol-earth-syst-sci-discuss.net/hess-2016-49/hess-2016-49-RC2- supplement.pdf

[AUTHORS' RESPONSE] We are grateful to the Reviewer #2 for offering us the remarks in the supplement. They will be conscientiously considered when we are allowed to work towards a revised manuscript.

---

## Author Response (AR1)

29 June 2016

Professor Helena Mitasova
Editor for the Special Issue of Hydrology and Earth System Sciences
"Geomorphometry: advances in technologies and methods for Earth system sciences"
(NHESS/HESS inter-journal SI)

Dear Editor,

I hereby submit the revised manuscript entitled "Observing river stages using unmanned aerial vehicles" (hess-2016-49) by Tomasz Niedzielski, Matylda Witek and Waldemar Spallek.

The detailed responses to comments offered by the Reviewers are submitted below. The responses follow my previous responses published as interactive comments. I would like to express our thanks to the Referees for evaluating our work. I believe that their remarks led to a significant improvement of the manuscript, both in terms of scientific completeness and the presentation. I hope the revised version is suitable for publication in Hydrology and Earth System Sciences.

I would like to take this opportunity and thank you very much for handling the manuscript.

Yours sincerely
Tomasz Niedzielski, Ph.D., Dr hab.
Professor at the University of Wrocław, Poland

**Responses to Reviewers' comments**

The shortened versions of the Reviewers' remarks are provided in brackets and typed using bold. The responses are fully provided, along the lines of our responses published in the interactive discussion (please note that certain repetitions occur when a given problem is raised by both Referees). The bullet points help to identify places in the revised (annotated) manuscript where key modifications have been made. The annotated manuscript is located in this document, below responses to Reviewers' comments.

1. **Responses to comments offered by the Reviewer #1**

   (a) **(General comment on needs for clarifying the manuscript and explaining methods)** We thank the Reviewer #1 for assessing that the manuscript presents reasonable results. Having read the reviews offered by two Referees we entirely agree that the manuscript should be substantially modified so that some parts on methods are better explained. The revised version of the manuscript clarifies many aspects, and uses new material (numerical example and table) to explain methods in detail.

(b) **(Need for parameters of all UAV flights)** The five flight missions have been performed with the comparable parameters. We double checked the UAV log files and confirmed that heights (both planned and measured during the missions), which determine the ground resolution, were kept approximately at a similar level.

- (Tab. 1) We included a new table (Tab.1 – please note that numbering of the subsequent tables was modified after incorporating a new Tab. 1).
- (2.1. Study area, fifth paragraph) We wrote a few sentences which refer to the new Tab. 1 and emphasize similar heights of all UAV missions.

(c) **(Need for explaining manual georeferencing to the LIDAR data)** In order to response to this comment we firstly put an emphasis on our key assumption which may be formulated as follows: "presence of a potential shift between two spatial data sets does not cause meaningful changes in area of the considered objects" (this is expressed in line 5 on page 11 of our HESS Discussion Paper). For instance, if one replicates an orthophotomap and applies a translation vector to such a newly produced spatial data set, the same objects will reveal the same areas (no change in scale and rotation). To support this finding we refer to a recent paper by Mesas-Carrascosa et al. (2014) [Mesas-Carrascosa F.J., Notario-García M.D., Meroño de Larriva J.E., Sánchez de la Orden M., García-Ferrer A., 2014. Validation of measurements of land plot area using UAV imagery. International Journal of Applied Earth Observation and Geoinformation 33, 270–279]. These authors argue that "Other shortcomings include the lack of vertical adjustment of the aerial camera and the unknown or variable interior orientation of the camera. These factors affect point position accuracy but do not necessarily decrease the accuracy of area measurements".

- (References) We added the paper by Mesas-Carrascosa et al. (2014) to a list of references.
- (2.2. UAV data processing, second paragraph) We added a few sentences on a relation between area accuracy and point position accuracy.
- (4. Results and discussion, eight paragraph) While discussing the results, we added a few sentences to emphasize the importance of what Mesas-Carrascosa et al. (2014) claim.

We also described the spline-based procedure that fixes all orthophotomaps to a single LIDAR data. Our explanation reads as follows: "We identified characteristic features in the LIDAR digital terrain model (DTM) which were evenly distributed and possible to identify in the orthophotomap. These features comprise: crossings of bounds, crossings of drainage ditches, and centres of bridges or passages (crossings of streams and roads). More than 10 points were used to perform georeferencing, as the spline method requires. A spline function allowed us to precisely georeference the control points (i.e. the aforementioned mutual features) and transform raster data set with continuity and smoothness, such as the rubber sheeting method."

- (2.2. UAV data processing, first paragraph) We literally added the above-mentioned explanation to the manuscript.

(d) **(Correction of spelling in sentences on visible light images)** We agree that the sentence highlighted by the Reviewer #1 does not read well in the initial version of the manuscript. We propose to rephrase the sentence so that it reads as follows: "For observing water surface area, the use is made of the following satellite-acquired measurements: high-resolution visible light images or infrared images, passive microwave data and radar images.". We went through several research papers and double checked that the notion of "visible light images" is properly used in the revised proposition.

- (1. Introduction, fourth paragraph) The above-mentioned corrected sentence was included in the introduction.

(e) **(Need for information on the accuracy of expert-based digitizing water extents)** The Reviewer #1 pointed out an important problem of the accuracy of a manual digitization carried out under several conditions (lines 27–31 on page 6 and the subsequent part of Subsection 2.2 in our HESS Discussion Paper). One of the most important factors that may potentially constrain a digitization accuracy is related to vegetation. Mapping vegetation with UAVs becomes popular as a recent paper by Husson et al. (2014) shows [Husson E., Hagner O., Ecke F., 2014. Unmanned aircraft systems help to map aquatic vegetation. Applied Vegetation Science 17, 567–577]. These authors focus on delineating edges between water and non-submerged aquatic as well as riparian species. They write that "In practise delineation was done by hand on paper printouts" and "Vegetation mapping, i.e. digitizing the UAS orthoimages, was performed manually by a human interpreter in a GIS using ArcGIS software". Although we concentrate on a fluvial environment, the idea behind our manual expert-based digitization remains similar to what Husson et al. (2014) proposed. It is worth noting that our digitization was practically carried out by two experts (GIS specialist + fluvial geomorphologist). Given this introduction, we unequivocally reply that the procedure met the assumed criteria (this is attained by the expert-based digitization). We also believe that the accuracy of the produced water surface area is acceptable. However, it was our intention to include Fig. 4 and Fig. 5 which help the reader to identify potential sources of errors.

- (References) We added the paper by Husson et al. (2014) to a list of references.
- (2.2. UAV data processing, fourth paragraph) A new paragraph has been added about impact of manual expert-based digitization on the accuracy of the polygon generation procedure. We referred to the above-mentioned paper by Husson et al. (2014) to support our approach.
- (2.2. UAV data processing, fifth paragraph) The beginning of the next paragraph has been rephrased so that we clearly discriminate between the impacts of vegetation and morphology on the accuracy of estimating water surface areas.

(f) **(Correction of spelling in sentences on ground control points)** Yes, indeed, the "GSPs" is a typo and in the revised manuscript is replaced by "GCPs".

- (2.2. UAV data processing, first paragraph) The correction has been made in the last sentence of the first paragraph.

(g) **(Need for detailed information on why non-chronological multitem-poral data are used)** Being grateful for notifying that the formulation of the hypotheses H0 and H1 is unclear, we hereby clarify the issue. The two sample Student's t-test is used to test a null hypothesis (H0) that means of two samples are equal, but three alternative hypotheses (H1) are allowed. These three alternatives include: means of two samples are different, mean of the first sample is bigger than mean of the second sample, mean of the second sample is bigger than mean of the first sample. In the latter two alternatives, the order of samples is important and has impact on where rejection region is located. Knowing the aforementioned basics, we stated the research hypothesis H0 with its alternative H1 on purpose, in the way that rejection of the null hypothesis implies acceptance of the alternative one (and this unequivocally indicates which area is meaningfully bigger). To clarify the entire problem, we suggest to consider two combinations of $L(j)$ and $L(k)$ (notations are explained in our HESS Discussion Paper). Recall that we test if mean$[L(j)]$ = mean$[L(k)]$ with the alternative that mean$[L(j)]$ < mean$[L(k)]$.

**CASE 1 (based on real data)**

$j$ = '27/11/2012'

$k$ = '13/05/2013'

mean[L('27/11/2012')] = $-1.71727$

mean[L('13/05/2013')] = $-1.501393$

Arithmetically, mean[L('27/11/2012')] is smaller than mean[L('13/05/2013')]. This inequality has also been confirmed statistically (the Student's t-test) at the significant level of 0.01 (see newly-numbered Tab. 6, grey box indicates that the difference in means is statistically significant). Hence, in this case a subsequent episode (time step $k$) revealed meaningfully bigger water surface area than the preceding one (time step $j$).

**CASE 2 (based on artificially modified data – changed order of dates)**

$j$ = '13/05/2013'

$k$ = '27/11/2012'

mean[L('27/11/2012')] = $-1.71727$

mean[L('13/05/2013')] = $-1.501393$

Arithmetically, mean[L('13/05/2013')] is not smaller than mean[L('27/11/2012')]. If we test (mean$[L(j)]$ = mean$[L(k)]$ with alternative mean$[L(j)]$ < mean$[L(k)]$) we cannot reject the null hypothesis with the Student's t-test at the significance level of 0.01. Hence in this case a subsequent episode (time step $k$) does not reveal a meaningfully bigger water surface area than the preceding one (time step $j$).

- (3.2. Interpretation through numerical exercise, entire subsection) We produced a new subsection which includes the description of the aforementioned numerical exercise. In addition, a new Subsection 3.1 has been added in order to present the previously published details on methods.

Since the newly-numbered Tab. 6 juxtaposes all cases of the above type ($j$ does not equal to $k$), we removed the following phrase "(but in practice $j < k$)"

which appeared in lines 3–4 on page 9 of the initial version of the HESS Discussion Paper. The deletion of the sentence will make the conclusions straightforward. Along these lines, we removed "$k = 2, \ldots, 5$" from results and discussion (in fact we analyse all possible transitions, not only from state 1 to states $k = 2, \ldots, 5$).

- (3.1. Concept, second paragraph) The inequality which suggests the chronological order has been removed.
- (4. Results and discussion, third paragraph) The list $k = 2, \ldots, 5$ has been deleted.

(h) **(Explanation why non-chronological transitions between characteristic stages have been used)** As we explained above, the order of $L(j)$ and $L(k)$ matters and influences the final results, however it is not necessary that $j < k$. The stages and transitions listed on page 10 are examples of low, mean, intermediate and high water levels. They have been recorded by real UAV flights on different dates. We used the UAV-observed water surface areas as true data that represent the aforementioned stages. We believe that, for the analysis that aims to check the procedure proposed in this paper, the chronological order of transitions between stages is not important. Of course, we agree with the Reviewer #1 that it would be ideal to have the chronological set of transitions, however such a data set is not available. Thus, we mixed the order to check various potential combinations of transitions.

- (4. Results and discussion, sixth paragraph) A new paragraph has been added to the main section on results and discussions. The paragraph explains why we used non-chronological transitions.

(i) **(Need for juxtaposing our data in a table)** The areas, fractions and logarithms (hence all input data used for the analysis) have already been juxtaposed in Tab. 5 of the initial version of the manuscript. This table received no. 6 in the revised version of the manuscript (new Tab. 1 included).

2. **Responses to comments offered by the Reviewer #2**

(a) **(Need for correcting the usage of English language)** We accept the criticism and thank the Reviewer #2 for spotting linguistic problems. The text has been corrected according to the marked suggestions. In addition, the entire manuscript has been edited.

- (Entire manuscript) Numerous corrections have been made to improve the level of English language used in the paper. They are all clearly marked in the annotated version of the revised manuscript.

(b) **(Two requirements: stability of height parameters and channel properties)** We thank the Reviewer #2 for finding our concept of reconstructing river stages using UAV-taken photographs a useful practice. We accept the criticism and believe that the revised manuscript does the above-mentioned idea sufficient justice. Our detailed responses are the following.

Indeed, flight altitude (and the resulting ground resolution) influences the water surface area observations. Our flight characteristics were kept stable over the five observational campaigns, hence the resolution is also stable over the entire

experiment. The similar comment has been also offered by the Reviewer #1. We double checked the flight logs and confirmed the comparability of UAV height parameters. The statistics calculated from the log data are juxtaposed in a new table to which we refer in the revised manuscript.

- (Tab. 1) We included a new table (Tab.1 – please note that numbering of the subsequent tables was modified after incorporating a new Tab. 1).
- (2.1. Study area, fifth paragraph) We wrote a few sentences which refer to a new Tab. 1.

We are grateful to the Reviewer #2 for mentioning the influence of channel morphology, mainly the slope of banks, on the observation of water surface area with the UAV. We agree that such a relationship exists. We also entirely accept the comment that the results, prepared for a specific river in the SW Poland, are not transferable to other rivers with different cross-sectional parameters. In fact, this has been already pointed out in the context of the relationship between water surface areas and river stages by Smith (1997) who argues that "Until additional empirical rating curves relating inundation area to ground measurements of stage or discharge are made, it is difficult to assess their potential for extrapolation to other rivers of similar morphology. However, it seems likely that such curves will vary significantly between rivers and therefore must be constructed for each site." [Smith L.C, 1997. Satellite remote sensing of river inundation area, stage, and discharge: a review. Hydrological Processes 11, 1427–1439]. However, our approach is centred on a statistical analysis of water surface areas, not river stages themselves. In fact, we quantitatively infer on statistically meaningful changes in water surface area (this is a key part of our procedure) and only qualitatively, through the existence of a relationship between water surface areas and river stages published by Usachev (1983) [Usachev V.F., 1983. Evaluation of food plain inundations by remote sensing methods. In: Proceedings of the Hamburg Symposium, IAHS Publ. 145, 475–482], extrapolate our results into changes in river stages. We believe that our quantitative approach (recall that this concerns seeking changes in water surface areas in the orthophotomaps produced from the UAV-taken photographs) forms a general method that – under several conditions clearly identified in the manuscript – may be applied in other regions. However, the use of the approach to infer on river stages should be made with caution, since such an extrapolation requires a knowledge about the relationship between water surface areas and river stages (and the characteristics of this relation are vulnerable to sites-specific river morphology, especially bank slopes). The relation between water surface area and stage is quasi-linear for rivers (Usachev, 1983) and strongly linear for lakes (Xia et al., 2983) [Xia L., Shulin Z., Xianglian L., 1983. The application of Landsat imagery in the surveying of water resources of Dongting Lake. Proceedings of the Hamburg Symposium, IAHS Publ. 145, 483–489]. In the revised manuscript we write about the strength of the relationship. Since our motivation was to offer a new method for checking if UAV-based observations of water surface area may be suitable for implementing the HydroProg-FloodMap-UAV procedure (described in the introduction), we also refer to a recent paper on the performance of Hydro-Prog [Niedzielski T., Miziński B., 2016. Real-time hydrograph modelling in the upper Nysa Kłodzka river basin (SW Poland): a two-model hydrologic ensemble prediction approach. Stochastic Environmental Research Risk Assessment, DOI: 10.1007/s00477-016-1251-5].

- (4. Results and discussion, tenth paragraph) A new paragraph has been added to the main section on results and discussions. The paragraph focuses on the impact of channel slopes on estimates of water surface area.
- (References) We added the paper by Xia et al. (1983) to a list of references.
- (4. Results and discussion, fourth paragraph) We mentioned about the strength of the relationship between water surface area and water levels, referring to papers by Usachev (1983) and a newly cited paper by Xia et al. (1983).
- (References) We added the paper by Niedzielski and Miziński (2016) to a list of references.
- (1. Introduction, second paragraph) The paper by Niedzielski and Miziński (2016) has been cited to provide a reference for HydroProg and its performance.

(c) **(Need for explaining manual georeferencing with respect to the LI-DAR data)** Yes, we are aware of possible problems that may be associated with measuring the area of polygons that are generated on a basis of orthophotomaps produced without GCPs. We also know that a procedure for delineating boundaries of water surface area needs to be clarified. The similar remarks have been also offered by the Reviewer #1, and the responses below use the arguments which we raised when replying to the comments provided by the Referee #1. Our detailed explanation below focuses on: area computation when GCPs are unavailable and delineating edges of water surface areas.

In line 5 on page 11 of our HESS Discussion Paper we argue that a presence of a potential shift between two spatial data sets does not cause meaningful changes in area of the considered objects. The arguments that support this statement can be found in a recent paper by Mesas-Carrascosa et al. (2014) [Mesas-Carrascosa F.J., Notario-García M.D., Meroño de Larriva J.E., Sánchez de la Orden M., García-Ferrer A., 2014. Validation of measurements of land plot area using UAV imagery. International Journal of Applied Earth Observation and Geoinformation 33, 270–279]. These authors argue that "Other shortcomings include the lack of vertical adjustment of the aerial camera and the unknown or variable interior orientation of the camera. These factors affect point position accuracy but do not necessarily decrease the accuracy of area measurements".

- (References) We added the paper by Mesas-Carrascosa et al. (2014) to a list of references.
- (2.2. UAV data processing, second paragraph) We added a few sentences on a relation between area accuracy and point position accuracy.
- (4. Results and discussion, eight paragraph) While discussing the results, we added a few sentences to emphasize the importance of what Mesas-Carrascosa et al. (2014) claim.

Having justified a stability of area measurements in the case of smaller point position accuracy, i.e. also in the case of shift of orthophotomaps produced without GCPs, we hereby describe the spline-based procedure that fixes all orthophotomaps to a single LIDAR data. We identified characteristic features

in the LIDAR DTM which were evenly distributed and possible to identify in the orthophotomap. These features comprise: crossings of bounds, crossings of drainage ditches, and centres of bridges or passages (crossings of streams and roads). More than 10 points were used to perform georeferencing, as the spline method requires. A spline function allowed us to precisely georeference the control points (i.e. the aforementioned mutual features) and transform raster dataset with continuity and smoothness, such as the rubber sheeting method.

- (2.2. UAV data processing, first paragraph) We literally added the above-mentioned explanation to the revised manuscript.

In lines 27–31 on page 6 and the subsequent part of Subsection 2.2 in the HESS Discussion Paper we listed three criteria labelled as (1), (2) and (3). While the latter two issues are associated with GIS methods (the same scale should be kept when carrying out a vectorization procedure and a cartographic projection as well as reference system should be unified before measuring areas), the first one is strongly related to environmental factors, mainly vegetation. This first statement has been explicitly highlighted by the Reviewer #2 as an element that needs to be clarified. Mapping vegetation with UAVs becomes popular as a recent paper by Husson et al. (2014) shows [Husson E., Hagner O., Ecke F., 2014. Unmanned aircraft systems help to map aquatic vegetation. Applied Vegetation Science 17, 567–577]. These authors focus on delineating edges between water and non-submerged aquatic as well as riparian species. They write that "In practise delineation was done by hand on paper printouts" and "Vegetation mapping, i.e. digitizing the UAS orthoimages, was performed manually by a human interpreter in a GIS using ArcGIS software". Although we concentrate on a fluvial environment, the idea behind our manual expert-based vectorization remains similar to what Husson et al. (2014) propose. It is worth noting that our vectorization was practically carried out by two experts (GIS specialist + fluvial geomorphologist). Given this introduction, we unequivocally reply that the procedure met the assumed criteria (this is attained by the expert-based vectorization). We also believe that the accuracy of the produced water surface area is acceptable. However, it was our intention to include Fig. 4 and Fig. 5 which help the reader to identify potential sources of errors.

- (References) We added the paper by Husson et al. (2014) to a list of references.
- (2.2. UAV data processing, fourth paragraph) A new paragraph has been added about impact of manual expert-based digitization on the accuracy of the polygon generation procedure. We refer to the above-mentioned paper by Husson et al. (2014) to support our approach.
- (2.2. UAV data processing, fifth paragraph) The beginning of the next paragraph has been rephrased so that we clearly discriminate between the roles of vegetation and morphology in the accuracy of estimating water surface areas.

(d) **(Need for commenting on statistical independence)** All assumptions of the Student's t-test have been checked using: the Ljung-Box test (independence), the Shapiro-Wilk test (normality), the D'Agostino test (symmetry as

a feature of the Gaussian distribution), the Anscombe- Glynn test (mesokurti-city as a feature of the Gaussian distribution). The tests, performed with the significance level of 0.01, suggest that each sample (we analyze 5 samples corresponding to five dates) is "internally" independent and normally distributed. In particular, the Ljung-Box test provides arguments for independence since p-values are equal to 0.059, 0.092, 0.444, 0.713, 0.828 (Tab. 3 in our HESS Discussion Paper, which is equivalent to Tab. 4 in the revised manuscript), for five consecutive dates. Hence, from a definition of statistical independence we infer that they cannot reveal autocorrelation. We would like to take this opportunity and put an emphasis on spatial independence which has not been investigated in our work. In addition, variances of paired data sets have been found to be similar (Tab. 4 in the HESS Discussion Paper, which is Tab. 5 in the revised manuscript).

- (4. Results and discussion, second paragraph) Four new sentences have been added to clarify the issue of "internal" independence which implies lack of autocorrelation.

(e) **(Need for the ANOVA test)** Indeed, the ANOVA test is a generalization of the t-test to more than two groups. However, although we process five samples our intention is to allow a pairwise comparisons. In other words, our approach is targeted at solving a simple operational problem: we have two sets of UAV-acquired observations carried out on two different dates, and would like to know if water surface area increased in comparison to the preceding observation. In our manuscript we simply have five observations, and this allows us to carry out many tests to make the inference more evident. However, the pairwise comparison is a fundamental feature of our procedure.

- (3.1. Concept, third paragraph) We explicitly wrote that we aim to carry out the pairwise comparison.

(f) **(Need for shortening the description of the study area)** We entirely agree with the Reviewer #2 that the description of the study area is too detailed.

- (2.1. Study area) General description of the study area was meaningfully shortened. In the present form, it focuses on geomorphological aspects which are relevant for understanding the paper. All unnecessary fragments have been deleted.
- (From 2.1.1. to 2.1.9.) All sub-subsections have been removed.

(g) **(Need for considering linguistic remarks offered by the Reviewer #2 as a supplementary file)** We are grateful to the Reviewer #2 for offering us the remarks in the supplement. They have been conscientiously considered in a revised manuscript.

**Observing river stages using unmanned aerial vehicles**

Tomasz Niedzielski[1], Matylda Witek[1], and Waldemar Spallek[1]

[1]Department of Geoinformatics and Cartography, Institute of Geography and Regional Development, Faculty of Earth Science and Environmental Management, University of Wrocław, pl. Uniwersytecki 1, 50-137 Wrocław, Poland

*Correspondence to:* Tomasz Niedzielski (tomasz.niedzielski@uwr.edu.pl)

**Abstract.** We elaborated a new method for observing water surface areas and river stages using unmanned aerial vehicles (UAVs). It is based on processing multitemporal [c1]five orthophotomaps produced from the UAV-taken [c2]visible light images of [c3]nine sites of the river, acquired with a sufficient overlap in each part. Water surface areas are calculated in the first place, and subsequently expressed as fractions of total areas of water-covered terrain at a given site of the river recorded on [c4]five

5 dates. The logarithms of the fractions are later calculated, producing [c5]five samples of size [c6]nine. In order to detect statistically significant increments of water surface areas between two orthophotomaps we apply the asymptotic and bootstrapped versions of the Student's t-test, preceded by other tests that aim to check model assumptions. The procedure is applied to five orthophotomaps covering nine sites of the Ścinawka river (SW Poland). The data have been acquired during the experimental campaign, at which flight settings were kept unchanged over nearly [c7]three years (2012–2014). We have found that it is possible

10 to detect transitions between water surface areas [c8]associated with all characteristic water levels (low, mean, intermediate and high stages). In addition, we infer that the identified transitions hold for characteristic river stages as well. In the experiment we detected all increments of water level: (1) from low stages to: mean, intermediate and high stages; (2) from mean stages to: intermediate and high stages; (3) from intermediate stages to high stages. Potential applications of the elaborated method include verification of hydrodynamic models and the associated predictions of high flows [c9] as well as monitoring water levels

15 of rivers in ungauged basins.

**1 Introduction**

A key problem in assessing [c10]performance of distributed [c11]hydrodynamic models, which predict water depth across a river channel and [c12]can therefore be used to simulate flood extent, is access to up-to-date information on true inundation. There are numerous approaches used to carry out such observations [c13]of inundation. They include: terrestrial observations of flood

[revised manuscript text omitted]

**3 Methods**

**3.1 Concept**

[c4]

Let us assume that we have $m$ UAV-based orthophotomaps that consist of observations of the same part of river channel carried out at times $t_1, \ldots, t_n$. Let us consider only such fragments of the orthophotomaps which meet the criteria outlined in

c20  which

c21

c22

c23

c1

c2

c3

c4 Section on "Methods" has been divided into subsections to clearly identify new part on how the outputs should be

[revised manuscript text omitted]

[c4]In order to clarify the formulation of the hypotheses H0 and H1 the following numerical exercise is proposed. Let us consider two combinations of $L(j)$ [c5]and $L(k)$. [c6]Recall that we test if means of $L(j)$ [c7]and $L(k)$ [c8]are equal, with the alternative that the mean of $L(j)$ [c9]is smaller than the mean of $L(k)$. [c10]Let us consider two cases based on data borrowed from Tab. 2.

25  [c11]CASE 1 [c12]

Let us assume that

$j = $ '27/11/2012',

$k = $ '13/05/2013',

mean of $L($'27/11/2012'$)] -1.71727$,

30  mean of $L($'13/05/2013'$)] = -1.501393$.

[c3] Text added.

[c1] what is often the case,

[c2] Section on "Methods" has been divided into subsections to clearly identify new part on how the outputs should be interpreted (new Subsection "Interpretation through numerical exercise").

[c3] Text added.

[c4] Text added.

[c5] Text added.

[c6] Text added.

[c7] Text added.

[c8] Text added.

[c9] Text added.

[c10] Text added.

[c11] Text added.

[c12] The

Arithmetically, mean of $L$('27/11/2012') is smaller than mean of $L$('13/05/2013'). This inequality has also been confirmed statistically (the bootstrapped Student's t-test) at the significant level of 0.01 (latter it is shown that the difference in means is statistically significant). Hence, in this case a subsequent episode (time step $k$) revealed meaningfully bigger water surface area than the preceding one (time step $j$).

5    [c1]CASE 2 [c2]

Let us unrealistically assume a reverse order of the above-mentioned numbers, namely

$j =$ '13/05/2013',

$k =$ '27/11/2012'.

Arithmetically, mean of $L$('13/05/2013') is not smaller than mean of $L$('27/11/2012'). If we test the null hypothesis (mean

10    of $L(j)$ is equal to mean of $L(k)$) against the alternative hypothesis (mean of $L(j)$ is smaller than mean of $L(k)$) we cannot reject the null hypothesis with the bootstrapped Student's t-test at the significance level of 0.01. Hence, in this case a subsequent episode (time step $k$) does not reveal a meaningfully bigger water surface area than the preceding one (time step $j$).

**4   Results and discussion**

It is apparent from Fig. 8 that fragments of sites S1–S9 are covered with water, the extent of which is dissimilar at different

15    dates of UAV observations. These dates correspond to low-, normal-, intermediate- and high-flow situations (see Subsection 2.3). Water surface areas in sites S1–S9 are juxtaposed in Tab. 3. Along the lines of Eqn. 1, Tab. 3 presents the ratios $R_i(j)$ and their logarithms $L_i(j)$ – for $i = 1, \ldots, 9$ and $j = 1, \ldots, 5$. The latter numbers become input data for the subsequent analysis.

Since we aim to compare five samples $L(1), \ldots, L(5)$, we first have to verify if they follow the i.i.d. structure. The p-values of the Ljung-Box, Shapiro-Wilk, D'Agostino and Anscombe-Glynn tests – juxtaposed in Tab. 4 – indicate that the five data

20    sets are trajectories of statistical samples (sequences of i.i.d. random variables) [c3]from the normal distribution. This can be inferred at the significance level of 0.013 or smaller. [c4]It is worth commenting here on statistical independence which should be understood in our exercise as an "internal" independence within each sample. Every sample is produced from areas of polygons spatially distributed along the river. In particular, the Ljung-Box test provides arguments for such an independence since p-values are equal to 0.059, 0.092, 0.444, 0.713, 0.828, for five consecutive dates. Hence, from a definition of statistical

25    independence we infer that they cannot reveal autocorrelation. In addition, variances between each pair of $L(1), \ldots, L(5)$ are shown to be similar at the significance level of 0.03 or smaller, as the Fisher's test suggests (Tab. 5). The statistical inference [c5] shows that the assumptions of the Student's t-tests are fulfilled.

Subsequently, we apply the Student's t-test to verify the above-mentioned hypothesis H0 against H1 (see Section 3), and we do so for each pair from $L(1), \ldots, L(5)$. The results are presented in Tab. 6 which juxtaposes p-values of the test, computed

30    as asymptotic and bootstrapped solutions. Gray background boxes indicate statistically significant differences in water surface areas, which suggests the rejection of the H0 hypothesis. This means that the mean water surface area is shown to be greater in the subsequent $L(k)$ sample[c6] than in the preceding one.

[c1] *Text added.*

[c2] The numbers below, borrowed from Tab. 2, are included in the revised version of the manuscript.

[c3]

[c4] *Text added.*

[c5]

[c6]  k=2,...,5,

It is known that water surface area is correlated with river [c7]stage. The characteristics of such relationships are reviewed by Smith (1997). Usually, the correlations are positive, [c8]quasi-linear for rivers (Usachev, 1983) [c9]and even strongly linear for lakes (Xia et al., 1983)[c10]. Hence, when water surface areas are analyzed in this paper in combination with river stages and their classes (Fig. 7) the following transitions are found to be meaningfully observable.

5   – Low stages (27/11/2012 and 21/08/2013) →

→ mean stage (27/09/2013),

→ intermediate stage (02/06/2014),

→ high stage (13/05/2013).

– Mean stage (27/09/2013) →

10   → intermediate stage (02/06/2014),

→ high stage (13/05/2013).

– Intermediate stage (02/06/2014) →

→ high stage (13/05/2013).

Noteworthy is the fact that the other transitions are found to be insignificant. To verify the adequacy of the detected changes
15   between the water surface areas, we again refer to Fig. 6 and Tab. 2 which present stages observed at the Gorzuchów gauge at the time of the UAV observations. The visual examination of the graphs and table [c1]indicates that no changes in water-covered areas correspond to no changes in river stages and, conversely, significant differences in water surface areas observed by the UAV at dissimilar times correspond to visually well seen changes of water levels. This inference allows us to positively verify the research hypothesis stated in this paper. Namely, even small changes in water surface areas are observable using the UAV
20   and – in addition – meaningful changes of river stages can also be [c2]inferred from the orthophotomaps based on the UAV-taken [c3]visible light photographs.

[c4]A remark should be given here about the non-chronological order of the above-mentioned transitions. These numbers serve as examples of low, mean, intermediate and high water levels. They have been recorded by a real UAV on different dates. We used the UAV-observed water surface areas as true data that represent the aforementioned stages. We believe that,
25   for the analysis that aims to check the procedure proposed in this paper, the chronological order of transitions between stages is not important. It would be ideal to have the chronological set of transitions, however such a data set is not available for our experiment. Thus, we mixed the order to check various potential combinations of transitions in order to test the procedure on the real UAV-acquired data.

A note should be given on the accuracy of the elaborated approach. We believe that potential sources of error may reside
30   in: (1) the SfM accuracy, (2) application of the SfM without GCPs, (3) problems with the determination of water boundaries due to presence of vegetation and undercuts. The quality of outputs from the SfM procedure depends on many factors – e.g. texture and light [c5]which influence a number of keypoints in every image – and hence not uncommonly we produce incomplete

[c7]
[c8]
[c9] *Text added.*
[c10] A reference by Xia et al(1983) has been added

[c1]

[c2]
[c3]
[c4] *Text added.*

[c5] *Text added.*

[revised manuscript text omitted]

[c1] The new reference – the paper by Husson et al. (2014) was mentioned in the response.

[c2] The new reference – the paper by Mesas-Carrascosa et al. (2014) was mentioned in the response.

[c3] The new reference – the paper by Niedzielski and Miziński (2016) has just been published, and it provided fundamental basics of HydroProg.

[c4] The new reference – the paper by Xia et al. (1983) refers the relationship of water surface area and stages for lakes.

[revised manuscript text omitted]